# FUNCTION INDUCTION AND TASK GENERALIZATION: AN INTERPRETABILITY STUDY WITH OFF-BY-ONE ADDITION

**Qinyuan Ye**[1,2]    **Robin Jia**[1]    **Xiang Ren**[1]
[1]University of Southern California    [2]Salesforce AI Research
{qinyuany, robinjia, xiangren}@usc.edu

## ABSTRACT

Large language models demonstrate the intriguing ability to perform unseen tasks via in-context learning. However, it remains unclear what mechanisms inside the model drive such task-level generalization. In this work, we approach this question through the lens of off-by-one addition (*i.e.*, 1+1=3, 2+2=5, 3+3=?), a two-step, counterfactual task with an unexpected +1 function as a second step. Leveraging circuit-style interpretability techniques such as path patching, we analyze the models' internal computations behind their performance and present three key findings. First, we identify a mechanism that explains the model's generalization from standard addition to off-by-one addition. It resembles the induction head mechanism described in prior work, yet operates at a higher level of abstraction; we therefore term it "function induction" in this work. Second, we show that the induction of the +1 function is governed by multiple attention heads in parallel, each of which emits a distinct piece of the +1 function. Finally, we find that this function induction mechanism is reused in a broader range of tasks, including synthetic tasks such as shifted multiple-choice QA and algorithmic tasks such as base-8 addition. Overall, our findings offer deeper insights into how reusable and composable structures within language models enable task-level generalization.[1]

## 1 INTRODUCTION

As the capabilities of language models (LMs) continue to grow, users apply them to increasingly challenging and diverse tasks, accompanied by evolving expectations (Zhao et al., 2024; Tamkin et al., 2024; Kwa et al., 2025). Consequently, it becomes impractical to include every task of interest in a model's training prior to deployment. In this context, task-level generalization—the ability of a model to perform novel tasks at inference time—becomes highly crucial and valued.

Prior work shows that LMs already exhibit this capability to a significant extent through in-context learning (Brown et al., 2020; Chen et al., 2022; Min et al., 2022a). The underlying mechanisms of this behavior are being actively investigated, with work on induction heads (Olsson et al., 2022) and function vectors (Hendel et al., 2023; Todd et al., 2024) offering substantial insights on pattern matching tasks (*i.e.*, [A][B]...[A] → [B]) and mapping-style tasks (*e.g.*, France: Paris, Australia: → Canberra). However, our understanding is still limited, especially regarding more complex generalization scenarios involving multi-step reasoning or newly-defined concepts in the task.

In this work, we aim to enhance our understanding of how models handle novelty and unconventionality with one counterfactual task: off-by-one addition (*i.e.*, 1+1=3, 2+2=5, 3+3=?). For humans, this task consists of two sequential steps: standard addition, followed by an unexpected increment of one to the sum. When a language model is prompted to perform this task with in-context learning, we anticipate two possible outcomes: (1) the model acquires the intended +1 operation and thus outputs 7, or (2) it adheres to fundamental arithmetic rules and outputs 6.

We begin our study by evaluating six contemporary LMs on off-by-one addition. Our findings indicate that all evaluated models consistently demonstrate the first outcome, effectively leveraging in-context

---

[1]Code: ⓞ INK-USC/function-induction

examples; furthermore, performance increases consistently as more shots are used. Motivated by these observations, we seek a more comprehensive understanding of how models perform off-by-one addition, and in particular, the +1 step of the task. To this end, we employ mechanistic interpretability and path patching techniques (Wang et al., 2023), which enables us to trace the model's output logits to a specific set of attention heads and their interconnections responsible for +1 behavior.

Our analysis with `Gemma-2 (9B)` (Gemma Team, 2024) reveals that the model's computation of +1 is mainly mediated by a circuit involving three groups of attention heads. Notably, two of these groups and their connections resemble the structure of the induction head mechanism described in prior work (Olsson et al., 2022)[2]. This observation leads us to hypothesize that the circuit implements a case of *function induction*—inductive reasoning that transcends *token-level* pattern matching and operates at the *function-level*. Our analysis also reveals that the +1 function is transmitted along six (or more) paths in the model's computation graph; in each path, an attention head writes a distinct fraction of the function, whose aggregate effect yields the complete +1 function.

We further validate the universality of our findings across models and tasks (Olah et al., 2020; Merullo et al., 2024). Regarding models, we repeat our analysis on `Mistral-v0.1 (7B)` (Jiang et al., 2023), `Llama-2 (7B)` (Touvron et al., 2023) and `Llama-3 (8B)` (Grattafiori et al., 2024), confirming the existence of the function induction mechanism, though in slightly varied forms. Regarding tasks, we extend our analysis with four task pairs—off-by-$k$ addition, shifted multiple-choice QA, Caesar Cipher, and base-8 addition—designed to replace sub-steps in off-by-one addition with substantially different operations. We demonstrate the reuse of the same mechanism in these task pairs.

Overall, our results advance our understanding of important language model capabilities such as in-context learning and latent multi-step reasoning. They highlight the flexible and composable nature of the function induction mechanism we have characterized, and provide substantive insights into how models may generalize when encountering novel task variations.

## 2 LMs Learn Off-by-One Addition in Context

Off-by-one addition is a synthetic, counterfactual task involving two steps. The first step is standard addition, and the second, unexpected step is a +1 function. In this work, we are interested in whether and how the model can perform this task with in-context learning. We provide concrete 4-shot examples of standard addition and off-by-one addition in Table 1. In this section, we first evaluate contemporary language models on this task and describe our observations.

| Base Task | Standard Addition | 4+3=7\n3+2=5\n6+0=6\n3+3=6\n1+0= **1** |
|---|---|---|
| Contrast Task | Off-by-One Addition | 4+3=8\n3+2=6\n6+0=7\n3+3=7\n1+0= **2** |

Table 1: **Example Prompt of Standard and Off-by-One Addition.** Red is used to mark the base prompt and answer. Orange is used to mark the contrast prompt and answer.

**Data.** To create the evaluation data, we randomly sample 100 test cases, each with 32 in-context examples ($a_i + b_i = c_i$) and one test example ($a_{test} + b_{test} = c_{test}$). We sample $a, b, c$ from the range of [0,999], and restrict that for all $i$, $c_{test} \neq c_i$. This is to make sure these test cases evaluate models on inducing +1 function, instead of copying and pasting the answer token ($c_{test}$) from the previous context ($c_i$).

**Models.** We evaluate six recent LMs on this task: `Llama-2 (7B)` (Touvron et al., 2023), `Mistral-v0.1 (7B)` (Jiang et al., 2023), `Gemma-2 (9B)` (Gemma Team et al., 2024), `Qwen-2.5 (7B)` (Yang et al., 2024a), `Llama-3 (8B)` (Grattafiori et al., 2024) and `Phi-4 (14B)` (Abdin et al., 2024). These models were developed by different organizations, employ different number tokenization methods, and were released in different years, thereby providing a diverse and representative sample. Please refer to Table 4 for details of these models.

---

[2] Induction heads (Olsson et al., 2022) facilitate a language model's token copying behavior in sequences like [A][B]...[A] → [B] by directly copying *token* [B] from the context. Our work aims to explain a more abstract, *function-level* behavior—how models induce the function $f(x) = x + 1$ from sequences like [A] $f$([B]) ... [C] → $f$([D]) (*e.g.*, 1+1 = 3 ... 3+3 = 7 ). See §6 and §A for further details.

**Evaluation Results.** In Fig. 1, we report the accuracy when different numbers of in-context examples are used. All evaluated models exhibit non-trivial performance on this task, demonstrating that this behavior is pervasive. Additionally, performance always improves as the number of shots increases, indicating effective utilization of the in-context examples. Notably, more recent models like `Llama-3 (8B)` and `Phi-4 (14B)` achieve the strongest performance, with near perfect results in the 8-shot experiments. More details of our evaluation (*e.g.*, reporting accuracy on standard addition, using a smaller number range like [0,9], or removing the restriction of $c_{test} \neq c_i$) are deferred to §B.

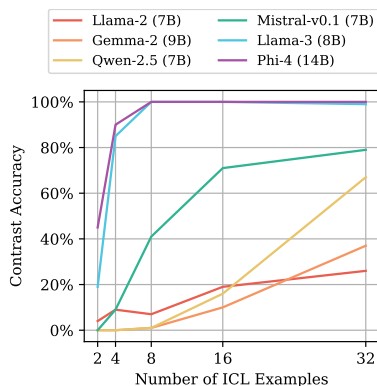

Figure 1: **In-context Learning Performance of Off-by-One Addition.**

## 3 Interpreting the Off-by-One Addition Algorithm

Off-by-one addition is likely unseen or highly underrepresented in the pre-training data, yet as Fig. 1 shows, contemporary language models can effectively induce the +1 operation with in-context learning. Intrigued by these observations, we aim to interpret the model's internal computation behind this behavior. §3.1 provides a brief overview of mechanistic interpretability and path patching, a line of methods that we find highly suited to our investigation. We further formalize our notation in this section. In §3.2 we describe our circuit discovery process and findings.

We choose `Gemma-2 (9B)` as the default model based on our preliminary experiments (§B), and use "1+1=3\n2+2=5\n3+3=?" as a running example in the following. Unless specified otherwise, all experiments below use 100 off-by-one addition test cases using numbers in the range of [0,9].[3]

### 3.1 Background: Mechanistic Interpretability and Path Patching

Mechanistic interpretability is a subfield of interpretability that aims to reverse-engineer model computations and establish "correspondence between model computation and human-understandable concepts." (Wang et al., 2023) A transformer-based language model can be viewed as a computation graph $M$, where components like attention heads and MLP layers serve as nodes, and their connections as edges. We use $M(y|x)$ to denote the logit of token $y$ when using $x$ as the input prompt. A circuit $C$ is a subgraph of $M$ that is responsible for a certain behavior. In our study, the behavior of interest is the induction and application of the +1 function in off-by-one addition.

The specific method we rely on is path patching (Wang et al., 2023), which is built on activation patching (Meng et al., 2022) and causal mediation (Vig et al., 2020) methods from prior work. In the past, such technique has supported interpretability findings on a wide range of model behaviors (*e.g.*, Hanna et al., 2023; Stolfo et al., 2023; Prakash et al., 2024b; Li et al., 2025).

Extending path patching to our case, we first run forward passes on both the base prompt $x_{base}$ (1+1=2\n2+2=4\n3+3=) and contrast prompt $x_{cont}$ (1+1=3\n2+2=5\n3+3=), to obtain the logits $M(.|x_{base})$ and $M(.|x_{cont})$. We will then (1) replace part of the activations in $M(.|x_{cont})$ with the corresponding activations in $M(.|x_{base})$; (2) let the replaced activations propagate to designated target nodes (*e.g.*, output logits, query of a specific head) in the graph; (3) replace the activations of the target nodes in $M(.|x_{cont})$ with the activations obtained in (2). The computation graph after such replacement is denoted as $M'$. If such a replacement alters the model's output of "3+3=**7**" back to "3+3=**6**", we would believe that the part has contributed to the computation of the +1 function.

To simplify the notation, we define $F(C, x)$ as the logit difference between $y_{base}$ (**6**) and $y_{cont}$ (**7**) when prompted with $x$ and using the circuit $C$ while knocking out nodes outside $C$ in the computation graph, *i.e.*, $F(C, x) = C(y_{base}|x) - C(y_{cont}|x)$. Following Wang et al. (2023), we quantify the effect of a replacement by first computing $F(M', x_{cont})$, and then normalize it by the logit difference

---

[3]To accommodate our computational resources, circuit discovery experiments (§3.2) were conducted with 4 shots (accuracy=33%), while circuit analysis experiments (§4) were performed with 16 shots (accuracy=86%).

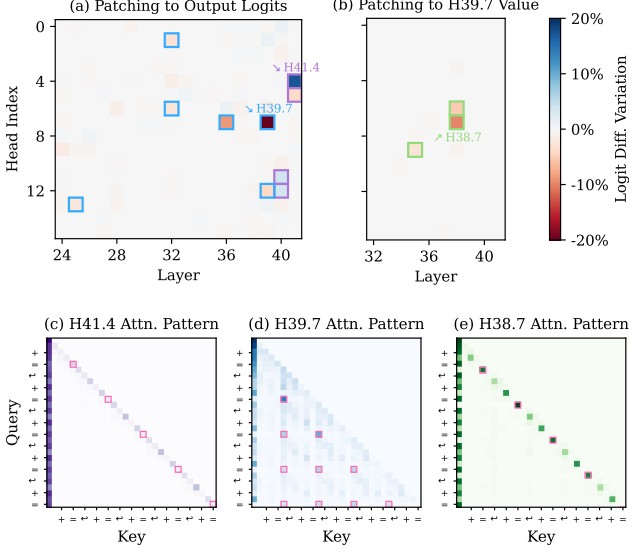

Figure 2: **Circuit Discovery with Gemma-2 (9B). Top: Patching Results on Selected Target Nodes.**
**(a)** We identify Group 1 heads and Group 2 heads that directly influence the output logits.
**(b)** We identify Group 3 heads that write to the value of H39.7.
**Bottom: Attention Pattern of Selected Heads.** We use 4 ICL examples in the format of "a+b=c\n". Causally-relevant positions are marked in pink.
**(c)** Group 1 heads mainly attend to the current token and <bos>.
**(d)** Group 2 heads attend to the answer tokens ($c_i$) of previous ICL examples at the position of "=".
**(e)** Group 3 heads attend to the preceding "=" at the position of $c_i$.

before intervention, *i.e.*, $r = \frac{F(M', x_{cont}) - F(M, x_{cont})}{F(M, x_{cont}) - F(M, x_{base})}$. See §C.1 for its expansion and explanations. The resulting ratio $r$, which we refer to as relative logit difference, will typically fall in the range of [-100%, 0%], with -100% representing the model favors $y_{base}$ (*i.e.*, the model losts its ability on off-by-one addition after replacement), and 0% representing the model favors $y_{cont}$.

## 3.2 CIRCUIT DISCOVERY

**Patching to the Output Logits.** Our investigation begins by setting the output logits as the target node, effectively asking "which attention heads directly influence the model output?" The results, visualized in Fig. 2(a), highlight 10 attention heads with a relative logit difference $|r| > 2\%$. [4]

We further investigate the attention pattern of the highlighted heads and categorized them into two groups. Group 1 heads appear exclusively in the last two layers of the model, and mainly attend to the current token and the <bos> token at each position (Fig. 2(c)).[5] Group 2 heads present periodical patterns consistent with the ICL examples in the prompt (Fig. 2(d)). Specifically, at the position of the last "=" token, where the model is expected to generate the answer as the next token, these attention heads will attend to the answer tokens ($c_i$) in previous ICL examples ($a_i + b_i = c_i$).

We additionally conduct path patching using the value of Group 1 heads as the target node, revealing that Group 2 heads also write to the value of Group 1 heads which then influence the final output logits. Combining these findings, we hypothesize that Group 1 heads are responsible for finalizing and aggregating information, while Group 2 heads are responsible for carrying the +1 function from the in-context examples to the test example.

**Patching to the Value of Group 2 Heads.** To further trace down the origin of the +1 function, we set the value of each head in Group 2 as the target node for path patching. For example, H39.7 (Head 7 in Layer 39) is a representative head in Group 2 with a relative logit difference $r$ of $-27\%$ when patching to the final output. When setting H39.7's value as the target node and performing path patching, three heads are highlighted (Fig. 2(b)) and all of these heads follow the pattern of attending to the previous token at certain positions (Fig. 2(e)). In particular, at the answer token $c_i$ in each in-context example, these head attend to the "=" token immediately before $c_i$. We repeat this procedure for remaining heads in Group 2 and identify more attention heads with the previous-token attending behavior. We collectively refer to them as Group 3 heads.

Our subsequent path patching attempts do not uncover any new attention heads leading to significant logit differences, thus we conclude the algorithm at this point.

---

[4]See §D.1 for additional analysis using smaller threshold values.

[5]The <bos>-attending behavior is commonly interpreted as a no-op and is prevalent in transformer-based language models (Barbero et al., 2025). See §C.3 for extended discussion.

**The Function Induction Hypothesis.** Fig. 3 provides an overview of the circuit we identified, illustrating the connections of the three head groups and highlighting the token positions they operate on. The comprehensive list of heads in each group can be found in §C.2.1 and Fig. 22(b).

We find it particularly intriguing that the structure of the circuit, in particular Group 2 and Group 3, resembles the structure of induction heads (Olsson et al., 2022), a known mechanism responsible for language model's copy-paste behavior. In the induction head mechanism, a previous token head "copies information from the previous token to the next token", and an induction head "uses that information to find tokens preceded by the present token." (Olsson et al., 2022)

Comparing these two mechanisms, induction heads could be seen as inducing a constant (zeroth-order) function $f$=output([B]), whereas in our mechanism, a first-order function $f(x) = x + 1$ is induced. Based on this intuition, the three groups of attention heads cooperate as follows:

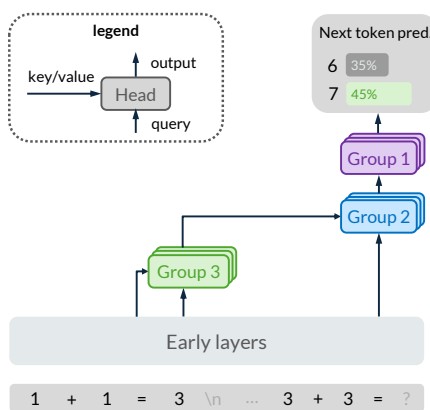

- Within an ICL example, at the "=" token (*e.g.*, "1+1="), the model initially drafts its answer via early-layer computations (*e.g.*, "2"), and anticipates to generate it as the subsequent token. However, at the answer token position $c_i$, the model encounters an unexpected answer (*e.g.*, "3"). Consequently, heads in Group 3 register this discrepancy at the position of $c_i$. Given their previous-token attending behavior, we name heads in Group 3 as previous token (PT) heads.

Figure 3: **Overview of the Identified Circuit.**

- In the test example portion of the prompt (*e.g.*, "3+3="), Group 2 heads retrieve the information registered by Group 3 heads at the "=" token, and subsequently writes out the +1 function. We name Group 2 heads as function induction (FI) heads as their operation resembles that of standard induction heads but applies to arithmetic functions rather than tokens.

- Lastly, we refer to Group 1 heads as consolidation heads, hypothesizing their role in finalizing the next-token output by synthesizing information from various sources.

## 4   CIRCUIT VALIDATION AND ANALYSIS

Previously, we constructed the function induction hypothesis based on our path patching results and its structural similarity to that of the induction heads mechanism. In this section, we dive deeper into the identified circuit, aiming to provide a more granular understanding.

**Initial Validation: Ablating FI Heads.** We begin our investigation with head ablation, a common technique to validate a head's involvement in a specific model behavior (Halawi et al., 2023; Wu et al., 2025). Here, we focus on FI heads and "ablate" a head by replacing its output in the forward pass on $x_{cont}$ with the corresponding head output in the forward pass on $x_{base}$. As shown in Fig. 4(a), the complete, unablated model achieved an accuracy of 86% on 16-shot off-by-one addition. Upon ablating the six FI heads, the model's behavior switched back to standard addition, resulting in 100% accuracy on standard addition and 0% on off-by-one addition. For a controlled comparison, we also ablated six randomly selected heads;

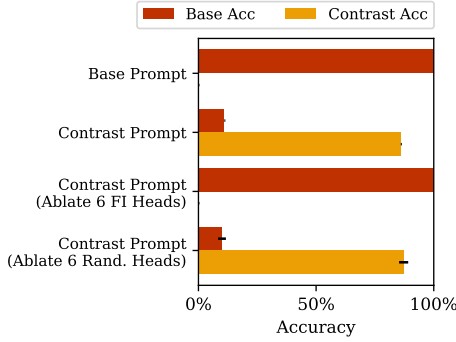

Figure 4: **Head Ablation Results.**

these showed minimal influence on either the base or contrast accuracy. This set of results provides preliminary evidence that the six FI heads are necessary in off-by-one addition.

**Further Validation: Measuring the Causal Effect of FI Heads.** In our hypothesis, FI heads are responsible for writing the +1 function to the residual stream at the "=" token. This behavior is highly

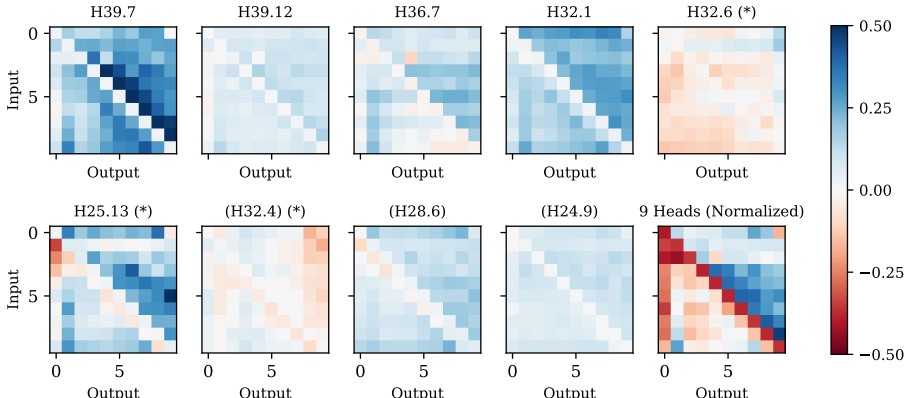

Figure 5: **Individual and Overall Effect of Identified FI Heads.** Each head writes out different information, which aggregates to implement the function of $f(x) = x + 1$ (bottom-right panel). (*) Effects of H32.6, H25.13, and H32.4 are rescaled to [-0.15, 0.15] to make the patterns more readable.

relevant to recent work (Todd et al., 2024; Hendel et al., 2023) which indicates that a small number of attention heads (*i.e.*, function vector heads; or FV heads) effectively transport task representations (*i.e.*, function vectors) in in-context learning. The FI heads we identified align with this description, and moreover, uncover a novel instantiation of the mechanism that operates within multi-step tasks.

The notion of function vectors inspires us to further validate the role of FI heads through their causal effect on a naive prompt $x_{naive}$, *e.g.*, "2=2\n3=?", for which the model is expected to assign a high probability to "3". If a FI head indeed writes out the +1 function, adding its output to the residual stream at the final "=" token should cause the model to increase its probability of generating "4".

Concretely, we construct the naive prompt "{x-1}={x-1}\n{x}=?" for $x \in [0, 9]$, and track the model's logits for tokens $[0, 9]$ both before and after adding the FI head output to the residual stream at the corresponding layer. This leads to a $10 \times 10$ heatmap, where the value at cell $(x_{input}, y_{output})$ represents the change in logits for token $y$ when the function vector is added.

In Fig. 5, we present these heatmaps for each of the six FI heads identified in §3.2. We include three additional heads (H32.4, H28.6, H24.9) that, while showing modest effects ($1\% < |r| < 2\%$) in §3.2, contribute meaningfully to the +1 function as revealed by this analysis. We find that FI heads work collaboratively—each of them contributes a distinct piece to the overall +1 function. For example, with an input $x$, H39.7 promotes $x + 1$, H28.6 suppress $x - 1$, H32.1 promotes digits greater than $x$, H24.9 suppresses $x$. When the outputs of these nine heads are added to the final residual stream altogether, their combined effect implements the +1 function, as depicted in the last panel of Fig. 5.

**Universality of Function Induction.** To investigate the universality of our findings across models, we repeat the path patching experiments with `Llama-3 (8B)`, `Llama-2 (7B)`, and `Mistral-v0.1` `(7B)`. We identified all three groups of heads across these models, except that the two consolidation heads identified in `Mistral-v0.1` display weaker and less consistent signals. Still, these observations provide promising evidence that the function induction mechanism we've found is general and consistently emerges across various language models. See §C.2 for more details.

**FI Heads (Ours) and FV Heads (Todd et al., 2024) are Two Disjoint Sets of Heads.** While our analysis demonstrates that FI heads transport task representations similarly to the FV heads described in prior work, a direct comparison with `Llama 2 (7B)` reveals important distinctions. Todd et al. (2024) reported that FV heads appear in early-middle layers of the model (before layer 20), whereas our FI heads are located in late layers of the model (layer 29-31). There is no overlap between the two sets of heads, suggesting that our work presents a distinct, previously undocumented finding. We hypothesize that FI heads can be seen as an instantiation of the broader FV head mechanism, but are only triggered in multi-step tasks when late layers are used to perform the late steps. See §C.2.4 for the full list of FI/FV heads and §6 for further discussion on their differences.

**Additional Analysis.** Due to space limits, we defer various supporting evidence to the appendix. We conduct a rigorous evaluation of our circuit using the *faithfulness*, *completeness*, and *minimality* criteria introduced in Wang et al. (2023). Our circuit mostly satisfies these criteria, and we discuss the results in §D. We deliberately focus on FI heads in §4 given the interesting insights from these results. We provide further validation and analysis of consolidation heads and previous token heads in §E.

## 5    TASK GENERALIZATION WITH FUNCTION INDUCTION

Our investigation so far suggests that function induction is the key mechanism enabling the model to generalize from standard addition and manage the unexpected +1 step in off-by-one addition. Given the importance of task generalization for capable AI systems, we aim to explore the broader usage of this mechanism. In this section, we investigate how the identified mechanism extends to a broader set of synthetic and algorithmic tasks. Specifically, §5.1 introduces the four task pairs examined, and §5.2 presents the overall findings and additional analyses for two of these pairs.

### 5.1    TASKS

| (a) Off-by-$k$ Addition | | (c) Caeser Cipher | |
|---|---|---|---|
| Standard | 4+3=7\n3+2=5\n6+0=6\n3+3=6\n1+0= **1** | ROT-0 | c -> c\nx -> x\ne -> e\nt -> t\nq -> **q** |
| Off-by-Two | 4+3=9\n3+2=7\n6+0=8\n3+3=8\n1+0= **3** | ROT-2 | c -> e\nx -> z\ne -> g\nt -> v\nq -> **s** |
| (b) Shifted MMLU | | (d) Base-$k$ Addition | |
| Standard | [...]\nAnswer: (B)\n[...]\nAnswer: **(A)** | Base-10 | 25+16=41\n60+16=76\n13+35=48\n52+17= **69** |
| Shift-by-One | [...]\nAnswer: (C)\n[...]\nAnswer: **(B)** | Base-8 | 25+16=43\n60+16=76\n13+35=50\n52+17= **71** |

Table 2: **Task Pairs Used in Task Generalization Experiments.** Red is used to mark the base prompt and answer. Orange is used to mark the contrast prompt and answer.

**(a) Off-by-$k$ Addition.** One extension of off-by-one addition is changing the offset to other values. Here, we consider offsets $k \in \{-2, -1, 2\}$. We use $k = 2$ as a representative case to be reported in the main paper. Results and analysis on the other offsets are deferred to §F.

**(b) Shifted Multiple-choice QA.** We consider going beyond arithmetic tasks and replace steps in off-by-one addition with substantively different steps. The base task is chosen to be multiple-choice QA questions on selected subjects of the MMLU dataset (Hendrycks et al., 2021). The contrast task is created with an additional step to shift the answer choice letter by one letter, *e.g.*, A→B, B→C.

**(c) Caesar Cipher.** One realistic task that leverages shifting functions is Caesar Cipher. During encoding, a letter is replaced by the corresponding letter a fixed number of positions down the alphabet (Wikipedia contributors, 2025). This task is also commonly used to evaluate a language model's reasoning capabilities (Prabhakar et al., 2024). Here we consider single-character Ceaser Cipher with different offsets $k \in \{-12, -11, \dots, 0, \dots, 12, 13\}$. We use $k = 0$ as the base task, and $k = 2$ as the representative contrast task.

**(d) Base-$k$ Addition.** Lastly, we consider the task of base-$k$ addition, which was used by Wu et al. (2024) to assess the a model's memorization versus generalization. Prior work (Ye et al., 2024) suggests that LMs may formulate a shortcut solution for base-8 addition by interpreting it as "adding 22 to the sum" from in-context examples; our interpretability analysis helps further investigate this observation. We consider two digit base-10 addition as the base task, and base-$k$ addition as the contrast task, with $k \in \{6, 7, 8, 9\}$. We use $k = 8$ as a representative case in the main paper.

### 5.2    RESULTS AND ANALYSIS

**FI heads are reused in a wider range of tasks.** Using the four task pairs introduced above, we examine the role of the function induction mechanism we discover with head ablation experiments, similar to the one done in Fig. 4. We run forward passes on both the base task and the contrast task. We then replace the FI heads outputs in $M(.|x_{cont})$ forward pass with the corresponding head outputs in the $M(.|x_{base})$ forward pass.

We report results of the representative cases in Fig. 6. In all four task pairs, we first see a non-trivial performance on the contrast task, indicating effective generalization. Upon ablating the six FI heads, we observe a consistent trend: the model's contrast accuracy substantially decreases; the base accuracy increases and often returns to a level comparable to that achieved with the base prompt. These findings suggest that the mechanism identified with off-by-one addition is largely reused in

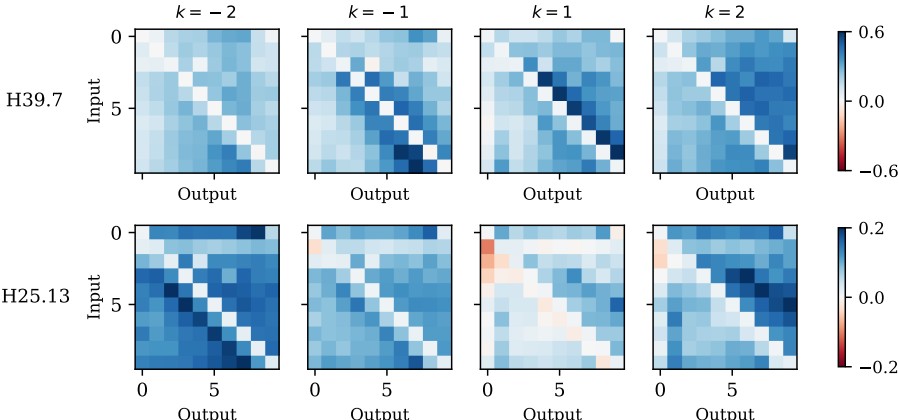

Figure 6: **Task Generalization with FI Heads.** In (d), base-8 addition has non-zero accuracy with the base-10 prompt, because in these test cases the base-10 answers happen to be correct in base-8.

Figure 7: **Effect of Two FI heads When Using Different Offsets in Off-by-$k$ Addition.**

these task pairs, which share a similar underlying structure but also represents substantially different sub-steps. This strongly demonstrates the mechanism's flexibility and composability.

We also observe that in (b) Shifted MMLU and (c) Caesar Cipher, the model has non-zero contrast accuracies when the FI heads are ablated. This implies that the six FI heads we found with off-by-one addition are useful, but not complete for these task pairs. See §F for additional discussion.

**Function vector analysis with off-by-$k$ addition.** We revisit the function vector style analysis done in Fig. 5, but this time considering different offsets $k \in \{-2, -1, 1, 2\}$. Results on two representative heads (H39.7 and H25.13) are shown in Fig. 7, with other heads deferred to Fig. 24-26.

We find that the effect of FI heads varies meaningfully with the offset $k$, demonstrating their generality and consistency with the hypothesized functionality. For the two selected heads in Fig. 7, we find that each of them has their own "specialty". For example, the heatmap for H25.13 suggests its primary responsibility for writing out $\pm 2$ functions. While its effect is stronger when the offset $k = \pm 2$, it still contributes in the case of $k = \pm 1$ by suppressing the original output $x$.

**Models struggle in base-8 addition due to under- or over-generalization.** It may sound unintuitive why the induction mechanism specialized in shifting functions could facilitate base-8 addition. One possible explanation is that the model initially performs standard base-10 addition with early layers, and apply minor adjustments when necessary. This adjustment step is possibly handled by the function induction mechanism in late layers.

Following this intuition, we propose one possible algorithm for two-digit base-8 addition in Listing 1. No adjustment is needed when there is no carrying over from the unit digit (Case 1), *e.g.*, $60 + 16 = 76$ is correct in both base-8 and base-10. When carry-over occurs, two separate cases needs to be considered. In Case 2, both the unit and the eight's place digit require adjustment, *e.g.*, $13_8 + 35_8 = 50_8$ and $13_{10} + 35_{10} = 48_{10}$, so both 4 and 8 in $48_{10}$ need to be adjusted. In Case 3, only the unit digit needs adjustment, *e.g.*, $25_8 + 16_8 = 43_8$ and $25_{10} + 16_{10} = 41_{10}$.

We randomly sample 100 32-shot prompts for each of these three cases, and track the model's behavior on the unit and eight's place digit. We report the results in Table 3. In Case 1, digits are adjusted unnecessarily in 7% (=6%+1%) of instances, suggesting over-generalization. Conversely,

in Case 2 and 3, digits were not adjusted as expected in 84% (=68%+16%) and 83% of instances, suggesting under-generalization. Overall, this evidence suggests that while the model can induce simple functions like +2 to some extent, it struggles with more complex situations where +2 should be only be triggered under certain *conditions*. Alternatively, if the induction of these conditions is viewed as an additional step in multi-step reasoning, the model we investigate may not yet be capable of two-step induction in a three-step task, thereby limiting their performance in base-8 addition.

```python
def base8addition(a, b):
    # (1) perform base-10 addition
    c = base10addition(a, b) # case 1
    # (2) apply adjustments
    if 8 <= a[0] + b[0] < 10: # case 2
        c[0] = (c[0] + 2) % 10
        c[1] = c[1] + 1
    elif a[0] + b[0] >= 10: # case 3
        c[0] = c[0] + 2
    return c
```

Listing 1: **One possible algorithm for two-digit base-8 addition.** This algorithm divides all scenarios into three cases. `c[0]` represents the unit digit and `c[1]` represents the tens/eights digit in a two-digit number `c`.

| Case | Full Model | | | | Ablate FI Heads |
| | Neither | c[0] | c[1] | Both | Neither |
| --- | --- | --- | --- | --- | --- |
| Case 1 | 93 | 6 | 1 | 0 | 100 |
| Case 2 | 68 | 0 | 16 | 16 | 100 |
| Case 3 | 83 | 14 | 0 | 0 | 100 |

Table 3: **Error analysis for two-digit base-8 addition.** We use 100 examples for each case specified in Listing 1. The correct behavior is marked in green . "Neither" suggests the number of times that neither `c[0]` or `c[1]` is adjusted, which is anticipated in Case 1. "`c[0]`" suggests that *only* `c[0]` is adjusted. "Both" suggests both digits are adjusted.

## 6 RELATED WORKS

**Mechanistic Interpretability.** The field of mechanistic interpretability aims to reverse-engineer complex neural networks into human-understandable algorithms (Bereska and Gavves, 2024; Sharkey et al., 2025), enhancing our understanding of a wide range of model behaviors, including long-context retrieval (Wu et al., 2025), and chain-of-thought reasoning (Cabannes et al., 2024). A common methodology involves analyzing their computation graphs of a specific task, as exemplified by studies on indirect object identification (Wang et al., 2023), "greater than" operation (Hanna et al., 2023), and entity tracking (Prakash et al., 2024a). Following this, our work begins with the off-by-one addition task, and showcases the broader applicability of our findings with various task pairs.

**Induction Heads in LMs.** Induction heads, described in Elhage et al. (2021) and Olsson et al. (2022), is a fundamental mechanism in language models that facilitate its pattern-matching behavior in sequences like [A][B]...[A] → [B]. This could be seen as inducing a *zeroth-order, constant* function $f$=output([B]), whereas our work identifies a circuit for inducing a *first-order, linear* function $f(x) = x + 1$, effectively generalizing the finding from token-level to function-level. Sharing our motivation of going beyond token-copying behavior, Minegishi et al. (2025) explored training two-layer transformers on carefully designed non-copying-based ICL tasks and investigated the circuit emergence. Ren et al. (2024) introduced the concept of semantic induction heads, which handles higher-level information processing such as syntactic dependencies and entity relationships in context.

**Function Vectors in LMs.** Recent work has characterized in-context learning in language models as the compression of in-context examples into a single task or function vector, which is subsequently transported to the test example to trigger the model to apply the function (Todd et al., 2024; Hendel et al., 2023; Yin and Steinhardt, 2025). These studies present strong evidence pertaining to *single-step, mapping-style* tasks like country-to-capital and English-French translation. Our work is inspired by this line of research, yet with two key differences: (1) We focus on off-by-one addition, a *multi-step arithmetic* task, where the learning of the second step depends on the results of the preceding step. (2) We provide a finer-grained interpretation on how function vectors, sent out by different attention heads, vary in content but collaborate to form a complete function. In concurrent work, this latter aspect was also explored by Hu et al. (2025), who investigate the task of add-$k$ (*i.e.*, "5→8, 1→4, 2→?") using subspace decomposition.

**Latent Multi-step Reasoning and Structural Compositionality in LMs.** Various studies investigate whether and how models perform latent multi-step reasoning, typically via multi-hop factoid QA tasks (Yang et al., 2024b; Wang et al., 2024). Our work demonstrates that LMs can dynamically infer the second step in a two-step problem from in-context examples, a process representing a novel, flexible and composable form of latent multi-step reasoning. More broadly, our findings are relevant

to research investigating structural compositionality (Lepori et al., 2023) (*i.e.*, breaking down complex tasks into subroutines) in language models.

# 7 CONCLUSION

In this work, we present an interpretability study on the off-by-one addition task, with the broader goal of investigating how language models handle unseen tasks using in-context learning. Our analysis uncovers a case of a function induction mechanism that captures the key "twist" required to generalize from seen to unseen tasks. This discovery extends and generalizes previous interpretability findings on induction heads and function vectors. We further show this mechanism is broadly reused beyond off-by-one addition, notably in realistic algorithmic tasks like Caesar Cipher and base-8 addition. Collectively, these observations deepen our understanding of what language models are capable of with in-context learning and multi-step reasoning, and how models generalize to novel tasks and situations. Moreover, our work provides compelling evidence that language models may develop composable and general mechanisms for handling ever-changing task variations, suggesting one possible pathway toward explaining and perhaps further enhancing model capabilities.

**Implications for LLM Development and Applications.** While our work focuses on very specific tasks and mechanisms, we believe the findings could further guide LLM development and application.

- **Evaluation:** In §5, we found that models achieve non-trivial accuracy on base-8 addition by relying on an unintended shortcut algorithm. This result strongly suggests that accuracy-based evaluation may disguise flawed reasoning processes inside the model. Complementing accuracy-based evaluation with interpretability analysis can help reveal the model's true capabilities and whether it has learned the intended reasoning process.

- **Pre-training:** Our analysis reveals that models perform multi-step reasoning latently and reuse a shared mechanism across tasks, a remarkable emergent structure given that these models are typically trained end-to-end on next-token prediction. This observation could inform the design of pre-training data mixtures or curricula that enhance multi-step compositional reasoning. For example, it may be beneficial to first train the model on single-step tasks (*e.g.*, standard addition) before exposing it to multi-step tasks (*e.g.*, off-by-one addition), thereby encouraging the development of function induction mechanisms.

- **Model Behavior and Alignment:** Recent work has identified concerning behaviors in language models, such as sycophancy (Sharma et al., 2025), agreement bias (Andrade et al., 2025), and susceptibility to belief shifts (Geng et al., 2025), which impact their reliability in real-world applications. We hypothesize that these behaviors may share structural similarities with the function induction mechanism we identified. Specifically, models may induce "belief-modifying functions" from context that drive their output generation. Investigating this connection could inspire future methods that better ensure reliability of language models.

**Future Work.** Beyond the discussion above, we believe the function induction mechanism itself is an intriguing interpretability finding that opens up many new research questions. In particular, investigating the emergence and formation of this mechanism during pre-training and attributing it to specific training instances could be an interesting future direction. For instance, one could test whether exposure to structurally related tasks (*e.g.*, puzzles or riddles) contributes to the emergence of the circuit. Additionally, Yin and Steinhardt (2025) discovered that function vector heads may have evolved from standard induction heads during pre-training. It would be interesting to investigate if such an evolutionary process applies to function induction heads in this work.

## LIMITATIONS

Regarding circuit discovery experiments (§3, §C and §D), the identified circuit is limited as it does not perfectly satisfy the faithfulness and completeness criteria, even with our best efforts. This challenge arises because achieving simultaneous satisfaction of faithfulness, completeness, and minimality is difficult, as these criteria often regulate each other. Moreover, number tokens are often mapped into a sinusoidal (Fourier) feature space rather than a linear space in language models (Nanda et al., 2023; Zhong et al., 2023; Zhou et al., 2024), which further complicates our interpretability analysis.

Regarding circuit analysis (§4 and §E), we mainly used causal intervention methods and examine the causal effect of attention heads on naive prompts. Future work could provide deeper mechanistic insights by analyzing the query-key and output-value circuits within these heads (Elhage et al., 2021), or by investigating the role of MLP layers in the overall mechanism.

Regarding task generalization experiments (§5 and §F), our current scope is limited to two-step tasks where the second step involves a shifting-related function. We view our results as identifying a *case* of function induction in language models, and we do not intend to claim a universal mechanism that applies to *any* function. That said, we anticipate that function induction, as a conceptual framework, may extend to a broader class of functions. Such behavior could be implemented by a shared circuit or by multiple specialized circuits; we leave a systematic investigation of this possibility to future work. Finally, the task pairs we investigated are synthetic or algorithmic; further exploration of the role of function induction heads on naturally occurring texts would be highly valuable.

## REPRODUCIBILITY STATEMENT

Our code and data are released at  `INK-USC/function-induction` and uploaded as supplementary material. Specifically, we include: (1) the datasets and code used to generate datasets; (2) the code for the main experiments reported in the paper; and (3) the code for creating the result figures. To further support reproducibility, we also provide environment configuration instructions and interactive notebooks that serve as a guided walkthrough. See §H for additional reproducibility details.

## ACKNOWLEDGMENT

We thank the anonymous reviewers for their helpful comments and feedback, which greatly strengthened our work. We thank Xisen Jin, Ting-Yun Chang, Lorena Yan, Yuqing Yang, Deqing Fu, Ollie Liu, Daniel Firebanks-Quevedo, Johnny Wei, as well as members of USC INK Lab and Allegro Lab for insightful discussions. QY and XR were supported in part by the Office of the Director of National Intelligence (ODNI), Intelligence Advanced Research Projects Activity (IARPA), via the HIATUS Program contract #2022-22072200006, the Defense Advanced Research Projects Agency with award HR00112220046, and NSF IIS 2048211. QY was also supported by a USC Annenberg Fellowship. RJ was supported in part by the National Science Foundation under Grant No. IIS-2403436. Any opinions, findings, and conclusions or recommendations expressed in this material are those of the author(s) and do not necessarily reflect the views of the National Science Foundation.

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

## A  INDUCTION HEAD MECHANISM AND FUNCTION INDUCTION MECHANISM

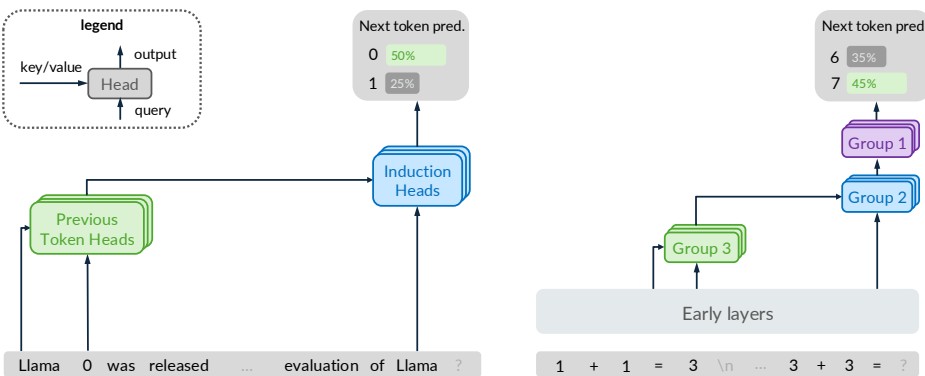

Figure 8: **Comparing Induction Head (Left) and Function Induction (Right).** See Fig. 9 for an annotated version of this figure.

**Comparing Induction Head and Function Induction.**    Fig. 8 provides a side-by-side visualization of the induction head mechanism (Olsson et al., 2022) and the hypothesized function induction mechanism (§3.2), demonstrating their structural similarity and explaining the basis for our hypothesis.

To provide a more concrete example on how induction heads work, consider the hypothetical scenario where a language model is completing the prompt: "Llama 0 was released in 2022. This paper presents an extensive evaluation of Llama ..." When the model first encounters an uncommon phrase, *e.g.*, "Llama 0", a previous token head will attend to "Llama" and register the information that "Llama appears before 0" at the position of "0". Later on, when "Llama" appears in the context again, an induction head will retrieve this piece of information from position of "0" and increase the likelihood of generating "0" as the next token. This induction head mechanism informs our hypothesis on function induction in §3.2 and the collaborative interaction between previous token heads and function induction heads in Fig. 8 (Right).

We also provide an additional figure (Fig. 9) that is annotated with the hypothesized roles of query, key, value and output representations.

**Relevance to In-context Learning with False Demonstrations.**    Various prior works investigate how language models handle false, random, or perturbed demonstrations in in-context learning (Min et al., 2022b; Yoo et al., 2022; Freeman et al., 2023; Wei et al., 2024; Lyu et al., 2023; Lin and Lee, 2024). Notably, Halawi et al. (2023) adopted an interpretability approach, observing the *overthinking* behavior of models (*i.e.*, models draft truthful answers at early layers and flip them to untruthful answers at late layers), and identified *false induction heads* that are responsible for copying the untruthful answers from the ICL examples.

Our analysis of off-by-one addition was largely motivated by these studies. Here we revisit the findings of Halawi et al. (2023) along with ours, using a unified view of two-step tasks, *i.e.*, $z = f(g(x))$. In Halawi et al. (2023), the first step, $y = g(x)$ is typically a text classification task, *e.g.*, news topic classification, and the second step, $z = f(y)$ is a permutation of the labels, *e.g.*, {Business→Sci/Tech, Sci/Tech→World, World→Sport, Sports→Business}. In our work, $y = g(x)$ is standard addition, and $z = f(y)$ is a +1 function.

In this view, our findings with off-by-one addition are consistent with those in Halawi et al. (2023), while also advancing the understanding in several aspects: (1) In both cases, language models decompose the task into two steps, and induce the second step based on the results of the first step. The second step could be either a conditional copy-paste function, *e.g.*, a permutation of labels, *or* an algorithmic function, *e.g.*, a +1 function. The latter represents a novel finding of this study, demonstrating that the second step can exhibit forms more complex than copy-paste operations. (2) Our path patching procedure has led us to identify two additional group of heads (consolidation heads and previous token heads) that are involved in handling false demonstrations. (3) Our work also suggests that the strategy to improve truthfulness by zeroing out false induction heads or function

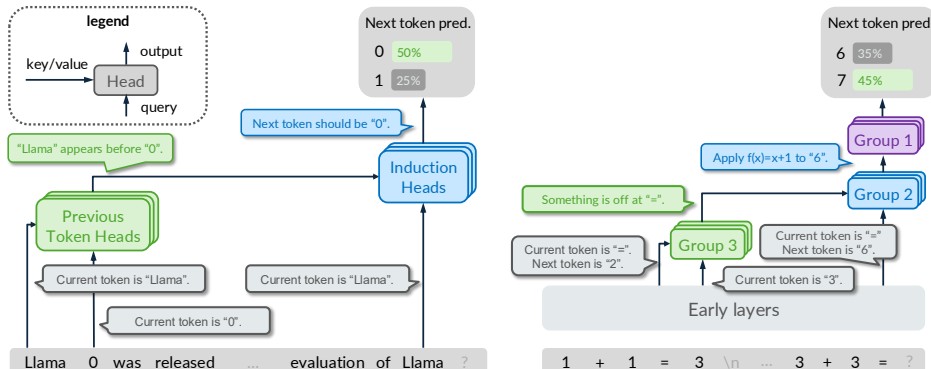

Figure 9: **Comparing Induction Head (Left) and Function Induction (Right).** In this figure, we've annotated the hypothesized roles of query, key, value, output representations of involved heads.

induction heads may have unintended consequences on models' positive capabilities, given their positive contributions to the cipher task and the base-8 addition task.

Related to the view of two-step tasks, Jain et al. (2024) demonstrate that models learn a "wrapper" function $g$ over an existing function $f$ in a sequential fine-tuning setting. Very recently, Yuan et al. (2025) show that models can learn to chain atomic functions into compositional functions during reinforcement learning. Our work and Halawi et al. (2023) suggest that language models demonstrate simple forms of this behavior with in-context learning as well.

**Relevance to Minegishi et al. (2025).** Highly relevant to our work and sharing the same motivation of investigating complex model behaviors in in-context learning, Minegishi et al. (2025) presents an in-depth study on training language models to perform in-context learning tasks and interpreting the mechanism. We discuss how our findings relate to theirs below.

In terms of the study design, both work study how transformer models perform in-context learning, using a *group* of task variants. Minegishi et al. (2025) designs a group of non-copying-based classification-style tasks, while we focus on algorithmic tasks. Additionally, Minegishi et al. (2025) trains small transformer models from scratch, enabling discovery of a three-phase circuit formation process. We instead interpret larger, off-the-shelf language models, which are more closely aligned with real-world applications and demonstrate strong capabilities across diverse tasks.

Regarding the findings, Minegishi et al. (2025) identifies a two-head circuit whose attention patterns and connections align with the previous token head and function induction head identified in our work. Both works find that models execute individual tasks through multiple parallel pathways and observe that models can adopt shortcut solutions for certain tasks by leveraging existing mechanisms. Our unique contributions are two-fold. First, we demonstrate that the mechanism we identify operates in two-step tasks, showing that models can perform latent multi-step reasoning. Second, we find that this mechanism is reused across many other tasks, suggesting broader compositional principles in model behavior.

## B    OFF-BY-ONE ADDITION EVALUATION

**Models.** In §2 we evaluated six recent language models on the task of off-by-one addition. In Table 4 we provide details of these models.

**Reporting base and contrast accuracy.** Previously in Fig. 1, we reported the accuracy of off-by-one addition (*i.e.*, the percentage of time that the model outputs 7 when given 3+3). In Fig. 10(a) we additionally report the accuracy of standard addition (*e.g.*, "3+3=6"), when the models are given the contrast prompt (*e.g.*, "1+1=3\n2+2=5"). We find that the base accuracy consistently decrease with more in-context learning examples. In Fig. 10(c), we show that models may also output numbers that are incorrect either in standard addition or off-by-one addition (*i.e.*, neither "6" or "7").

| Model Name | Huggingface Identifier | Reference | Tokenization 0-9 | 0-999 |
|---|---|---|---|---|
| Llama-2 (7B) | meta-llama/Llama-2-7b-hf | Touvron et al. (2023) | ✓ | |
| Mistral-v0.1 (7B) | mistralai/Mistral-7B-v0.1 | Jiang et al. (2023) | ✓ | |
| Gemma-2 (9B) | google/gemma-2-9b | Gemma Team (2024) | ✓ | |
| Qwen-2.5 (7B) | Qwen/Qwen2.5-7B | Yang et al. (2024a) | ✓ | |
| Llama-3 (8B) | meta-llama/Meta-Llama-3-8B | Grattafiori et al. (2024) | | ✓ |
| Phi-4 (14B) | microsoft/phi-4 | Abdin et al. (2024) | | ✓ |

Table 4: **Models Evaluated on Off-by-One Addition.** "0-9" means the model uses digit-level tokenization for numbers, *e.g.*, "123" is tokenized into ["1","2","3"], "0-999" means all numbers smaller than 1000 are considered one single token, *e.g.*, "123" is tokenized into ["123"].

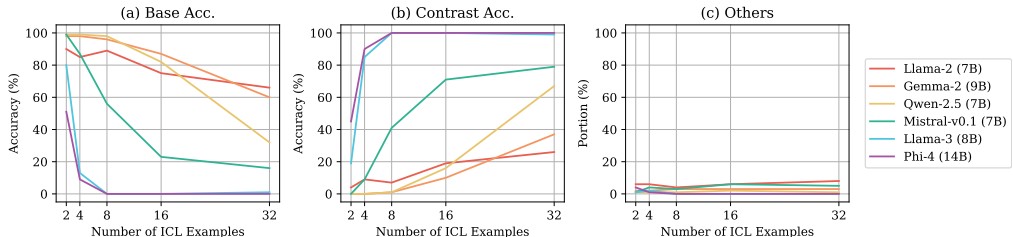

Figure 10: **Off-by-One Addition Evaluation, Reporting Base Accuracy.**

**Results in a smaller number range.** Previously in Fig. 1, we reported results when the operands were sampled from the range of [0,999]. In Fig. 11, we additionally report results when sampling from the range of [0,9] and [0,99]. For two models using 0-9 tokenization (Gemma-2 (9B) and Qwen-2.5 (7B)), the performance drops with larger number ranges. For the remaining models, the performance remains stable regardless of the number ranges.[6]

**Results with/without the constraint of $c_{test} \neq c_i$.** Previously in §2 we deliberately impose the constraint that $\forall i, c_{test} \neq c_i$. This is to rule out the possibility that language models perform off-by-one addition via copying $c_{test}$ from previous contexts. In Fig. 12, we compare the results of two additional sampling strategies: (1) no constraint on $c_{test}$ and $c_i$; (2) $\exists i, c_{test} = c_i$. By comparing Fig. 12(b) and (c) we see that for Mistral-v0.1 (7B) and Gemma-2 (9B), the accuracy is higher when $\exists i, c_{test} = c_i$. This observation implies that these two models leverages copy-paste induction more than function induction in performing off-by-one addition, though more rigorous analysis is required to draw a conclusion.

**Results with off-by-$k$ addition.** In Fig. 13-14, we present 32-shot off-by-$k$ addition results with various offsets $k$ using Gemma-2 (9B) and Llama-3 (8B) respectively.[7] One consistent trend is that models struggle more with offsets $k$ of larger absolute values. While Llama-3 (8B) generally outperforms Gemma-2 (9B), Gemma-2 (9B) demonstrates strong performance when $k = \pm 10$, potentially due to its adoption of 0-9 tokenization. An additional observation reveals that Gemma-2 (9B) typically achieves stronger performance with even values of $k$ compared to odd values.

---

[6]We chose Gemma-2 (9B) as the default model in our study because (1) we focused on the range of [0,9] in early stage of this work to prioritize simplicity, and Gemma-2 (9B) performs competitively in this setting; (2) Qwen-2.5 (7B), Llama-3 (8B), Phi-4 (14B) were not released or integrated into transformer-lens at that time. We acknowledge this experimental design limitation and address it by interpreting Llama-3 (8B) and Mistral-v0.1 (7B) in §C.

[7]The visualization of Fig. 13-14 was inspired by Prabhakar et al. (2024).

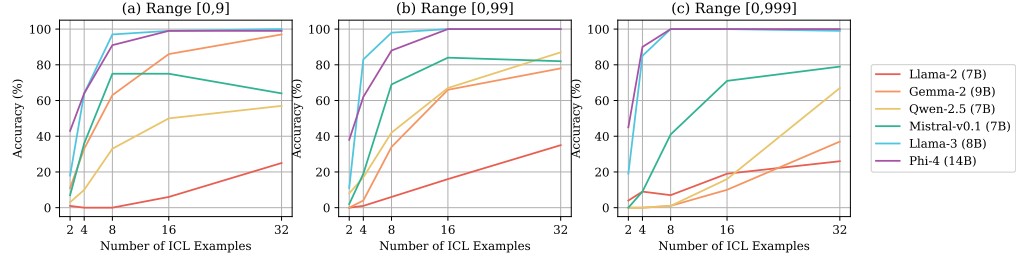

Figure 11: **Off-by-One Addition Evaluation, Using Smaller Number Ranges.**

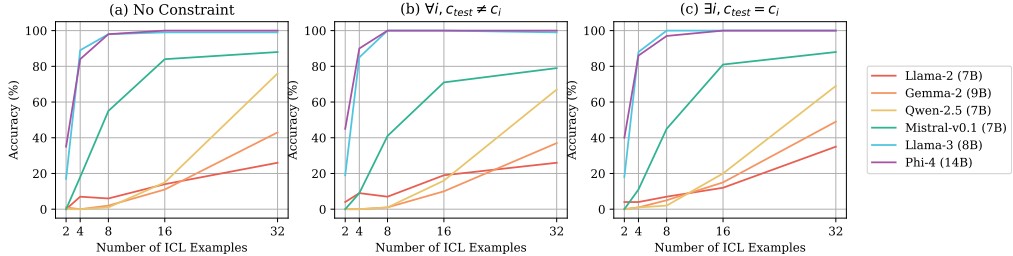

Figure 12: **Off-by-One Addition Evaluation, Different Sampling Constraints.**

## C  CIRCUIT DISCOVERY

### C.1  RELATIVE LOGIT DIFF

§3.1 introduced $r$, the relative logit difference, to measure the effect of a replacement during circuit discovery. We now elaborate on this formula to enhance clarity.

$$r = \frac{F(M', x_{cont}) - F(M, x_{cont})}{F(M, x_{cont}) - F(M, x_{base})} \tag{1}$$

$$= \frac{[M'(y_{base}|x_{cont}) - M'(y_{cont}|x_{cont})] - [M(y_{base}|x_{cont}) - M(y_{cont}|x_{cont})]}{[M(y_{base}|x_{cont}) - M(y_{cont}|x_{cont})] - [M(y_{base}|x_{base}) - M(y_{cont}|x_{base})]} \tag{2}$$

### C.2  IDENTIFIED HEADS

In the main paper, we focus on interpreting `Gemma-2 (9B)`. To explore the universality of the mechanism, we additionally conduct path patching with `Llama-3 (8B)`, `Llama-2 (7B)` and `Mistral-v0.1 (7B)`. We list the identified attention heads below.

#### C.2.1  `Gemma-2 (9B)`

`Gemma-2 (9B)` has 42 layers and 16 heads per layer. Path patching experiments were conducted with 4-shot off-by-one addition with numbers sampled from range [0,9].

- Consolidation Heads: H41.4, H41.5, H40.11, H40.12;
- Function Induction (FI) Heads: H39.7, H39.12, H36.7, H32.1, H32.6, H25.13;
- Previous Token (PT) Heads: H38.6, H38.7, H38.9, H35.14, H35.9, H31.4, H31.5, H29.5.

#### C.2.2  `Llama-3 (8B)`

`Llama-3 (8B)` has 32 layers and 32 heads per layer. Path patching experiments were conducted with 4-shot off-by-one addition with numbers sampled from range [0,999]. We visualize the path patching results in Fig. 15.

- Consolidation Heads: H31.1, H30.25, H29.11, H29.10, H28.16, H28.17, H28.18;

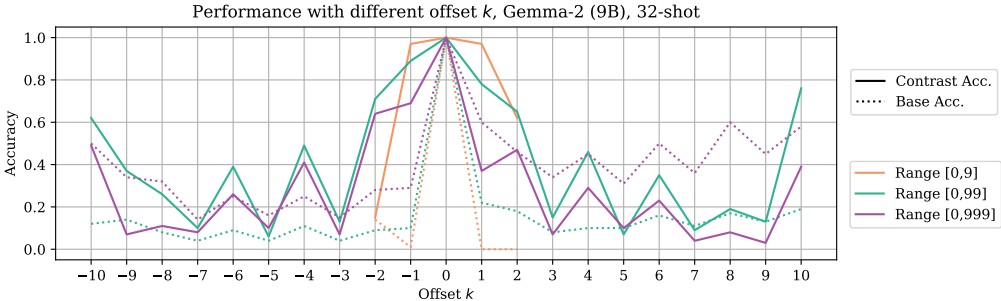

Figure 13: **Off-by-$k$ Addition Evaluation, `Gemma-2 (9B)`**

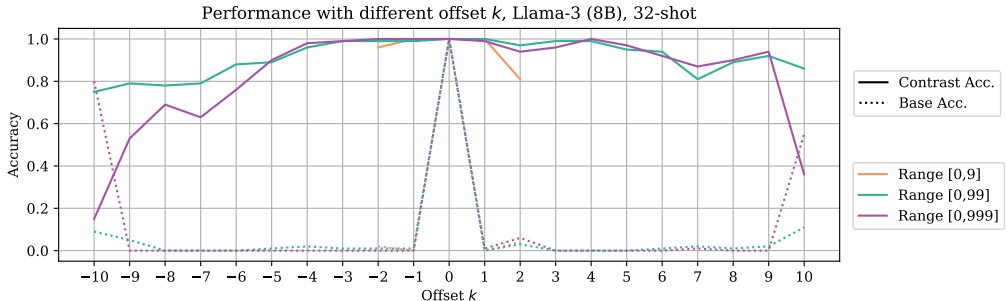

Figure 14: **Off-by-$k$ Addition Evaluation, `Llama-3 (8B)`**

- **Function Induction (FI) Heads**: H26.2, H23.13, H23.15;
- **Previous Token (PT) Heads**: H24.10, H24.11, H22.25, H22.27, H21.7.

### C.2.3 `Mistral-v0.1 (7B)`

`Mistral-v0.1 (7B)` has 32 layers and 32 heads per layer. Path patching experiments were conducted with 4-shot off-by-one addition with numbers sampled from range [0,9]. We visualize the results in Fig. 16. For the two consolidation heads in the list below, they show weaker effect and attend to both the current token and some other tokens, which slightly deviates from our findings with `Gemma-2 (9B)`. Apart from this, the results using `Mistral-v0.1` are consistent with other models.

- **Consolidation Heads**: (H31.10), (H31.1)
- **Function Induction (FI) Heads**: H30.2, H30.3, H30.4, H30.8, H30.10, H30.18, H31.2
- **Previous Token (PT) Heads**: H29.4, H29.6, H29.7.

### C.2.4 `Llama-2 (7B)`

`Llama-2 (7B)` has 32 layers and 32 heads per layer. Path patching experiments were conducted with 4-shot off-by-one addition with numbers sampled from range [0,9]. We visualize the results in Fig. 17.

**Two Variations.** All three groups of heads are present in `Llama-2 (7B)`. However, we notice two small variations compared to the circuit in `Gemma-2 (9B)`. **(1)** We identified H16.24 that achieves $r = 2.12\%$, but its attention pattern doesn't fit that of a consolidation head or a function induction head. Since $2.12\%$ is slightly above the $2\%$ threshold we set, we consider this noise; **(2)** One previous token head (H29.1) no longer attends to the "=" immediately before the answer token $c_i$ at the token $c_i$. Instead, it attends to the "=" token one ICL example away.

**Comparison with Todd et al. (2024).** To discuss our findings with those of Todd et al. (2024) with a common ground, we extract the list of FV heads by selecting the 10 heads with the highest absolute

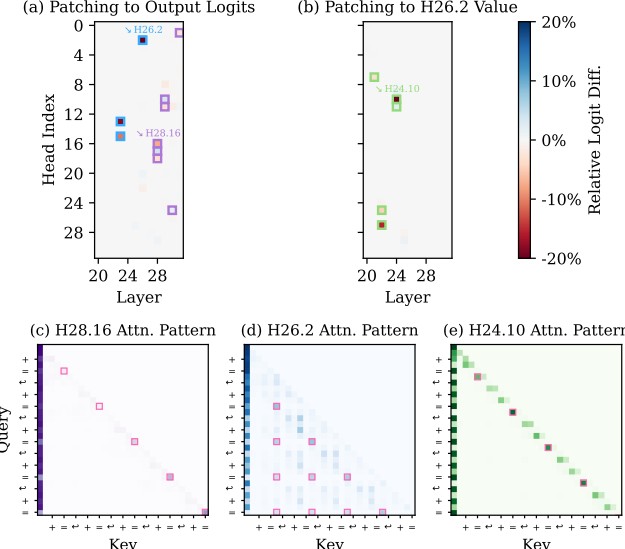

Figure 15: **Circuit Discovery with Llama-3 (8B).** Causally-relevant positions are marked in pink. Results are consistent with those with Gemma-2 (9B) in Fig. 2.

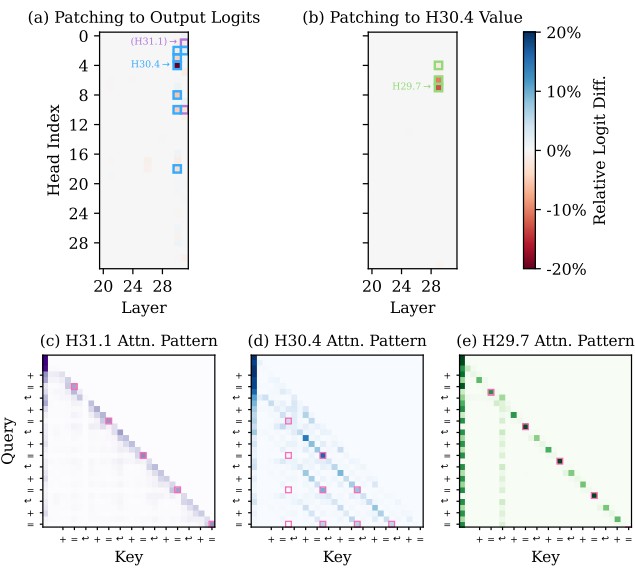

Figure 16: **Circuit Discovery with Mistral-v0.1 (7B).** Causally-relevant positions are marked in pink. Results are mostly consistent with those with Gemma-2 (9B) in Fig. 2, with the exception of the consolidation heads showing weaker signals.

average indirect effect (AIE) from Fig. 19 in Todd et al. (2024). These heads are concentrated in early-middle layers (before layer 20), whereas our FI heads appear in late layers (layers 29-31). There is no overlap between the two sets.

- Consolidation Heads: H31.28, H31.10, H30.3;

- Function Induction (FI) Heads: H31.30, H31.4, H29.26, H29.16, H30.26, H30.3;

- Previous Token (PT) Heads: H30.13, H29.1, H28.5, H28.10, H28.16, H28.24, H27.31;

- Miscellaneous Heads: H16.24;

- Function Vector Heads (Todd et al., 2024): H9.25, H11.2, H11.18, H12.15, H12.18, H12.28, H13.7, H14.1, H14.16, H16.10.

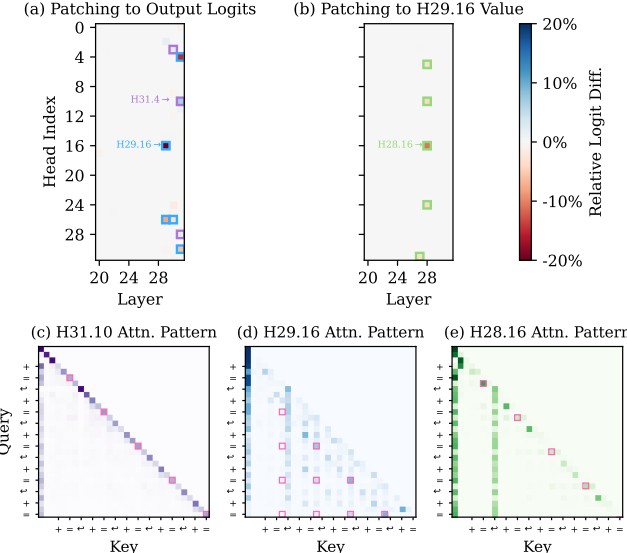

Figure 17: **Circuit Discovery with `Llama-2 (7B)`.** Results are mostly consistent with those with `Gemma-2 (9B)` in Fig. 2. Causally-relevant positions are marked in pink. In H29.16 and H28.16, the first ↩ token receives significant attention. This may represents an approximate "no-op", similar to typical <bos>-attending behavior in language models (§C.3).

## C.3 <BOS> ATTENDING BEHAVIOR OF IDENTIFIED HEADS

By visualizing the attention patterns of the heads in the function induction mechanism, we found that many heads attend to the <bos> token. In most cases, this happens at positions not causally relevant to our tasks, hence, we defer discussion here in the appendix.

Attending to <bos> is a prevalent behavior in language models. Barbero et al. (2025) showed that "almost 80% of the attention is concentrated on the <bos> token" in Llama-3 (405B). This phenomenon is sometimes referred to as "attention sink," (Xiao et al., 2024), and has attracted a lot of interest in the research community. A common interpretation in the literature is that attending to <bos> represents an approximate "no-op" or "resting" operation (Gu et al., 2025; Barbero et al., 2025; Clark et al., 2019; Vig and Belinkov, 2019). Since attention weights must sum to 1 due to the softmax operation, the model learns to attend to <bos> when attention to other tokens is not needed in the current context.

## C.4 ADDITIONAL INTERPRETABILITY ANALYSIS

### C.4.1 LOGIT LENS ANALYSIS

In this section, we apply logit lens (nostalgebraist, 2020), a widely-adopted interpretability method, to off-by-one addition. This involves directly computing the logits from intermediate layer representations using the final layer norm and the final unembedding layer.

We use `Gemma-2 (9B)` and 100 16-shot examples in this set of experiments. In Fig. 18, we report the logits of the base answer $y_{base}$ (*i.e.*, model outputting 3+3=**6**), the contrast answer $y_{cont}$ (*i.e.*, model outputting 3+3=**7**) and their differences, computed using the contrast prompt $x_{cont}$ (*i.e.*, 1+1=3) as model input. In Fig. 19, we repeat the experiments using $x_{base}$ (*i.e.*, 1+1=2) as the input prompt.

By comparing Fig. 18(a) and Fig. 19(a), we find that the curves in the two subplots begin to diverge notably after layer 25. This supports our claim that the model performs standard addition in the early layers and apply the +1 function in late layers.

Additionally, by comparing Fig. 18(b) and Fig. 19(b), we find that the logit diff decreases sharply after layer 38 in Fig. 18(b), a phenomenon absent in Fig. 19(b). This is consistent with our findings that H39.7 and H39.12 contribute significantly to writing out the +1 function to the residual stream.

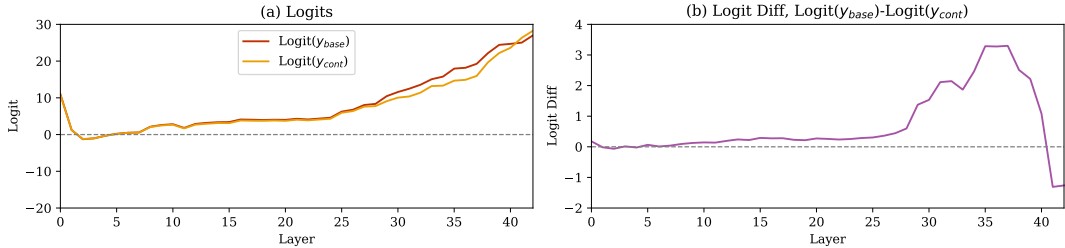

Figure 18: **Logit Lens Results when Using $x_{cont}$ as the Input Prompt.**

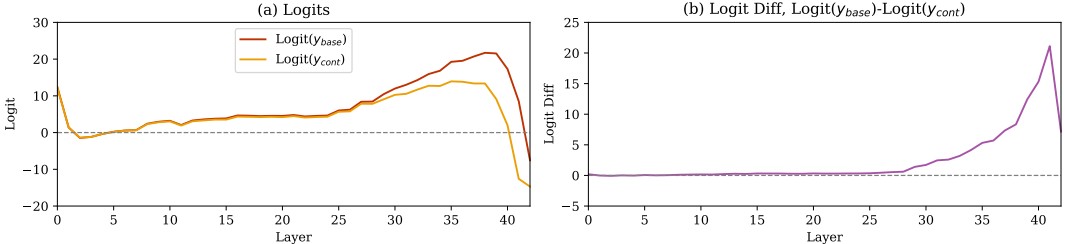

Figure 19: **Logit Lens Results when Using $x_{base}$ as the Input Prompt.**

### C.4.2 ACTIVATION PATCHING ANALYSIS

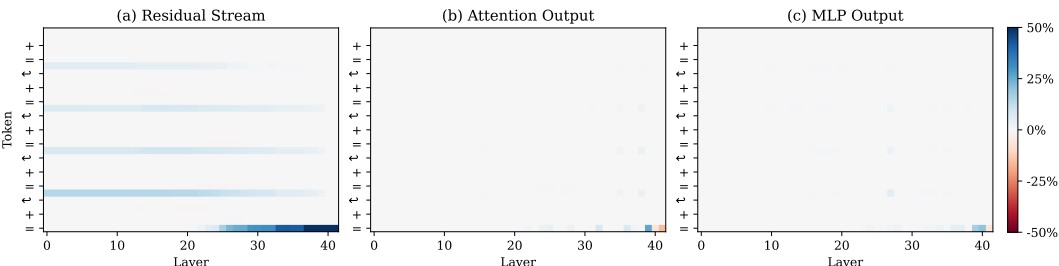

Figure 20: **Activation Patching By Token.**

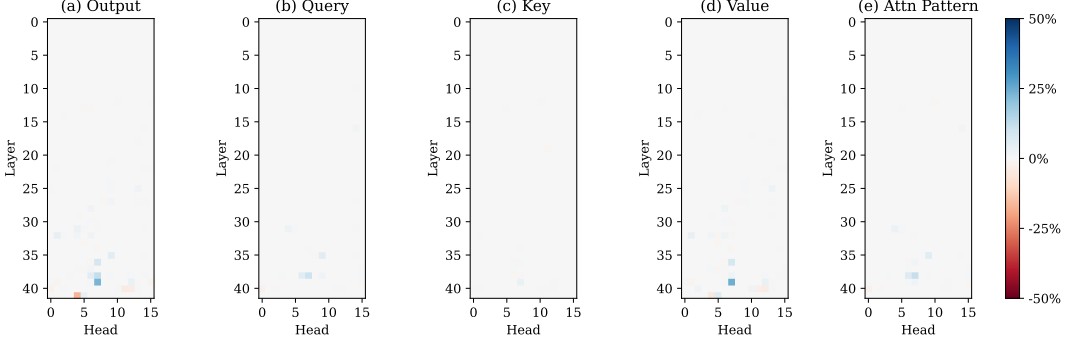

Figure 21: **Activation Patching By Head.**

In this section, we apply activation patching (Meng et al., 2022) to off-by-one addition. We performed this analysis in the early stages of our work to gather initial intuitions and signals for our problem, before transitioning to path patching for a more fine-grained understanding of the model's internal computation.

We use Gemma-2 (9B) and 100 4-shot examples in this set of experiments. First, we run forward passes for both the base prompt $x_{base}$ and the contrast prompt $x_{cont}$. We store the activations and

then replace the activations in the $x_{cont}$ forward pass with corresponding activations in the $x_{base}$ forward pass. We consider activation patching by token (Fig. 20) and by head (Fig. 21). We report the ratio $r' = 1 + r = \frac{F(M',x_{cont})-F(M,x_{base})}{F(M,x_{cont})-F(M,x_{base})}$ in these figures following previous works. We scaled the colormap in the figures to the range of [-50%, 50%] for clear visualization.

Fig. 20(a) visualizes the information flow from in-context examples to the residual stream of the last " =" token. Additionally, Figure 20(b) highlights several layers, specifically layers 32, 36, and 39 at the last "=" token, and layers 35 and 38 at the answer tokens $c_i$ in the in-context examples. This aligns with the FI heads (H36.7, H39.7, H39.12) and PT heads (H35.9, H35.14, H38.6, H38.7, H38.9) identified in §3.2. Figure 20(c) further reveals that MLP layers also play critical roles at certain positions. It is possible that FI heads write the +1 function to the residual stream, with subsequent attention and MLP layers involved in the execution of the +1 function. This hypothesis is inspired by prior observations of how MLP layers in transformer models are involved in arithmetic operations (Nanda et al., 2023; Stolfo et al., 2023). In this work, we limit our focus to attention heads, deferring further analysis of MLP layers to future work.

Results in Fig. 21 guide and complement our path patching experiments in §3.2. The identified PT heads (H35.9, H38.6, H38.7, H38.9) are highlighted in Fig. 21(b) and the FI heads (H36.7, H39.7, H39.12, H32.1, H25.13) are highlighted in Fig. 21(d).

### C.4.3 ALTERNATIVE HEAD ABLATION METHODS

In the main paper, we "ablate" or "knock out" a head by replacing its output in the $x_{cont}$ forward pass with the corresponding head output in the $x_{base}$ forward pass. This instance-specific ablation approach is adopted to better isolate the +1 function computation in each instance. However, this differs from ablation methods commonly used in interpretability work, such as zero ablation (Halawi et al., 2023) or mean ablation (Wang et al., 2023).

To demonstrate the consistency of our findings across different ablation settings, we repeated the experiment in Fig. 4 using zero ablation and mean ablation. For mean ablation, we averaged head outputs at the final "=" token from 100 standard addition examples. We found that in all ablation settings (zero ablation, mean ablation, and our instance-specific ablation), the contrast accuracy reduced to 0% and the base accuracy increased to 100% after ablation.

## D CIRCUIT EVALUATION

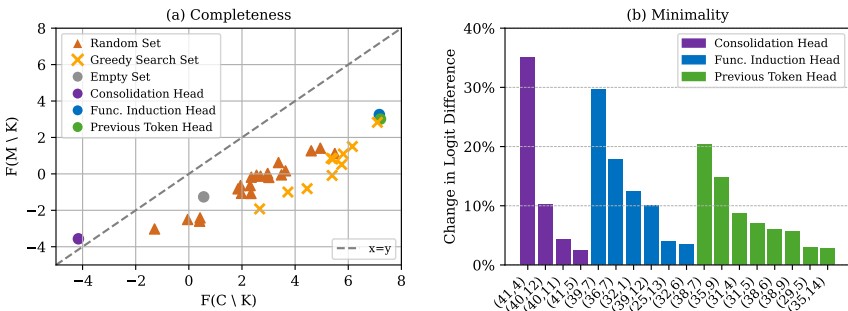

Figure 22: **Circuit Evaluation.**

In §4, we primarily validated the identified circuit using head ablation experiments and causal effect visualizations. Wang et al. (2023) proposed a more rigorous framework for circuit evaluation, based on **faithfulness**, **completeness**, and **minimality**. In the following, we evaluate the identified circuit according to these metrics. Note that we focus on interpreting the "off-by-one" component of the task, rather than the standard addition component. Hence, these circuit evaluation metrics are adapted accordingly to use $F(M, x_{base})$ as a reference point.

The **faithfulness** metric measures whether a circuit $C$ has a similar performance to the full model $M$, i.e., whether $F(C, x_{cont})$ is close to $F(M, x_{cont})$, with $F(C, x)$ defined earlier in §3.1. We find that

$F(M, x_{base}) = 7.17$, $F(M, x_{cont}) = -1.26$, and $F(C, x_{cont}) = 0.56$, suggesting that $C$ recovers $\frac{7.17 - 0.56}{7.17 - (-1.26)} = 78.4\%$ of the performance of $M$.

The **completeness** criterion evaluates whether for each subset $K \subseteq C$, the difference between $F(C \backslash K, x_{cont})$ and $F(M \backslash K, x_{cont})$ is small. In the following, we will omit the $x_{cont}$ term for brevity. We use various different sets (*e.g.*, randomly or greedily selected) as $K$ and report the results in Fig. 4(b). We find most points representing $(F(C \backslash K), F(M \backslash K))$ fall slightly below the $x = y$ line, while maintaining a monotonic trend, suggesting that the circuit $C$ is partially complete. This represents the best we can achieve with our current methodology. We also find that when $K$ is the set of all PT heads or all FI heads, both $f(C \backslash K)$ and $f(M \backslash K)$ are high, suggesting that the model favors $y_{base}$ in next-token generation (*i.e.*, 3+3=6) and switches back to standard addition under these ablation conditions. These observations are consistent with our function induction hypothesis.

Lastly, the **minimality** criterion measures whether each head $v$ in $C$ is necessary, by seeking a subset $K \subseteq C \backslash \{v\}$ that has a high score of $|F(C \backslash (K \cup \{v\})) - F(C \backslash K)|$. We manually constructed the $K$ sets for this purpose. As shown in Fig. 4(c), each head in $C$ is relevant to the task and has a non-trivial effect (>2%) in performing off-by-one addition.

### D.1 Improving the Identified Circuit with Local Search

Our circuit evaluation above is based on the circuit discovery procedure described in §3.2, where we used a 2% threshold to highlight relevant heads and ensure that the number of heads involved is manageable. This evaluation reveals that circuit $C$ achieves a faithfulness score of 78.4%, recovering a substantial portion of the model's behavior but suggesting we may have missed several heads responsible for the remaining 22%.

To address this, we employ a local search method. Specifically, we enumerate heads not in $C$ (*i.e.*, $v \in M \backslash C$), add each to $C$ to form a new circuit $C' = C \cup \{v\}$, and evaluate the faithfulness score of $C'$. We identify 7 heads that produce faithfulness score changes greater than 1.5%, which we report in Table 5. These include 6 heads (H37.6, H38.8, H40.0, H37.4, H24.9, H32.4) that increase the faithfulness score and 1 head (H36.6) that decreases it.

Including the 6 score-increasing heads in $C$ (denoted $C_6$) raises the faithfulness score to 89%. Including all 7 heads (denoted $C_7$) reduces the faithfulness score to 82%, which still exceeds the 78% achieved by the original circuit $C$.

We further examine the attention pattern of these heads. H37.6, H24.9, H32.4, and H36.6 exhibit patterns consistent with FI heads, while H38.8 fits the description of consolidation heads. However, H40.0 and H37.4 do not fit any head group in our function induction hypothesis. At "=" tokens, H40.0 attends to itself in the forward pass of $x_{base}$ but not in the forward pass of $x_{cont}$, likely suppressing the standard addition answer ($y_{base}$) by weakening its consolidation. H37.4 attends to "=" tokens in previous ICL examples at each "=", likely retrieving the consolidation signal from earlier ICL examples.

|  | $\Delta$ Faithfulness (%) | Rel. Logit Diff $r$ (%) |
|---|---|---|
| H37.6 | 3.13 | 0.88 |
| H38.8 | 2.33 | 0.55 |
| H40.0 | 2.29 | -1.26 |
| H37.4 | 1.96 | 0.39 |
| H24.9 | 1.86 | 1.64 |
| H32.4 | 1.67 | 1.29 |
| H36.6 | -3.44 | -1.15 |

Table 5: **Identifying Missing Heads with Local Search.** The "Rel. Logit Diff $r$" columns reports the relative effect $r$ of these heads when patching to the final logits (§3.2). Previously we set a threshold of 2% and hence these heads were not identified.

# E  CIRCUIT ANALYSIS

Due to space limit, we mainly perform circuit analysis on function induction (FI) heads in the `Gemma-2` (9B) model and present the most notable findings in the main paper (§4). In this section, we discuss remaining findings on FI heads in §E.1. We also present additional analysis on consolidation heads in §E.2 and previous token (PT) heads in §E.3.

## E.1  FUNCTION INDUCTION (FI) HEADS

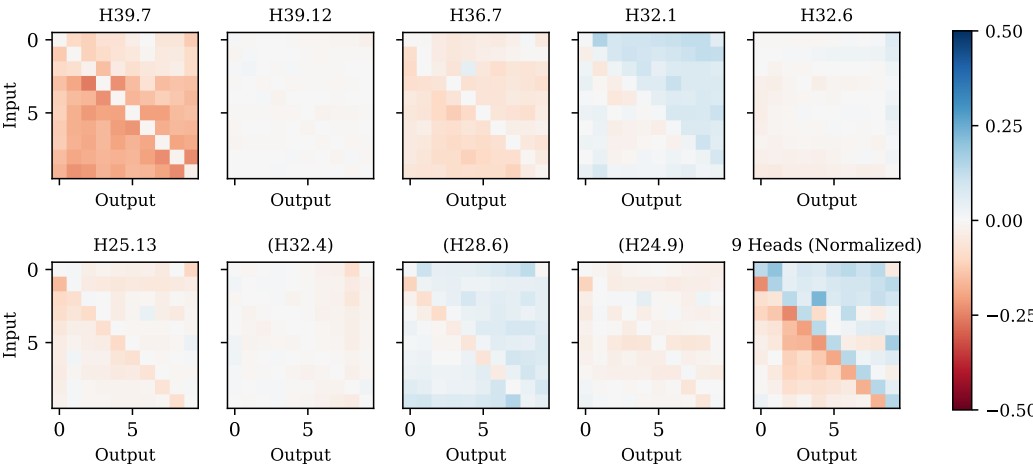

Figure 23: **Individual and Overall Effect of Identified FI Heads (Standard Addition).**

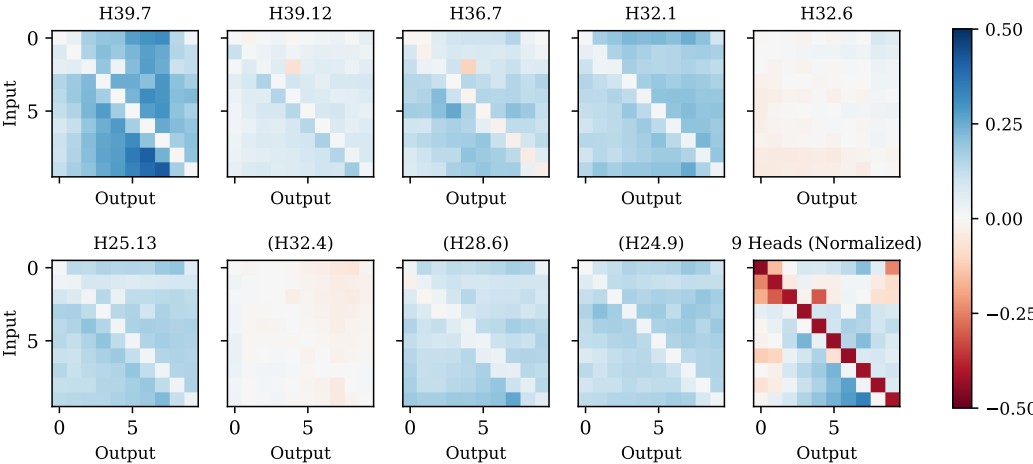

Figure 24: **Individual and Overall Effect of FI Heads in Off-by-$k$ Addition, $k = -2$.**

**What do FI heads write out in standard addition?**  Our function vector style analysis in §4 primarily focuses on what FI heads write out in off-by-one addition. However, these heads may also assume roles in standard addition. To investigate this, we add the FI head outputs in the $M(.|x_{base})$ to the naive prompt $x_{naive}$, and visualize the effect in Fig. 23. By comparing Fig. 5 and Fig. 23, we observe that most FI heads contribute meaningful but distinct information in standard addition, with H39.12 being an exception given its minimal effect in standard addition. The aggregated effect in the bottom-right panel in Fig. 23 suggests that FI heads collectively suppress $x - 1$ and promote $x$ in standard addition.

One possibility is that FI heads sharpen the answer $x$, or double-check it by performing $(x - 1) + 1$ in standard addition. In contrast, during off-by-one addition, the standard addition answers are first

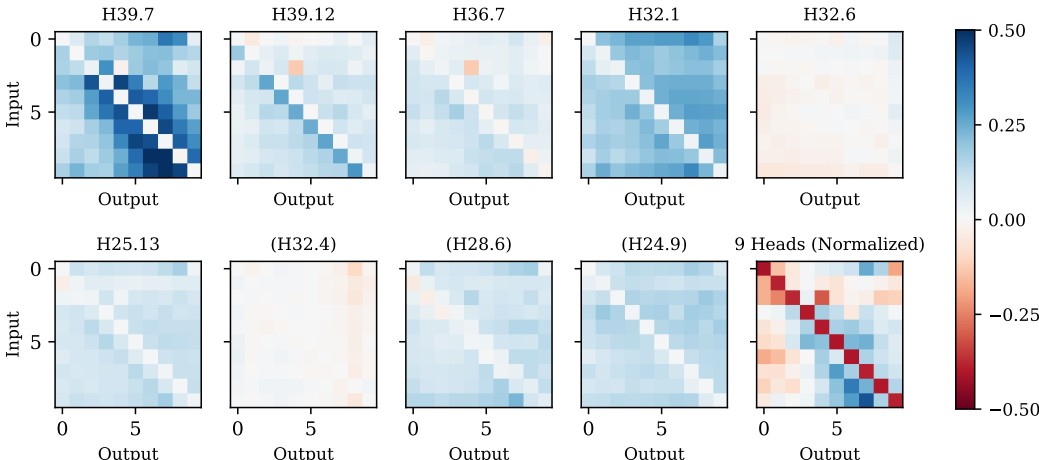

Figure 25: **Individual and Overall Effect of FI Heads in Off-by-$k$ Addition, $k = -1$.**

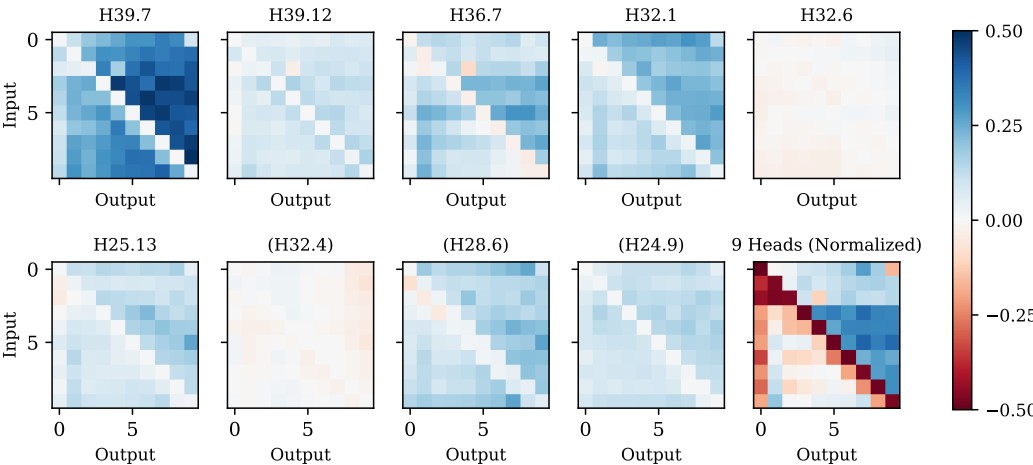

Figure 26: **Individual and Overall Effect of FI Heads in Off-by-$k$ Addition, $k = 2$.**

"locked in" after early layers, and the FI heads are repurposed to perform +1. We leave further investigation of this phenomenon to future work.

**What do FI heads write out in off-by-$k$ addition?** Previously in Fig. 7, we demonstrated how the effect of H39.7 and H25.13 changes with respect to different offset $k$. In Fig. 24-26 we report the effect of all nine heads when $k = -2, -1, 2$. We find that for some heads (*e.g.*, H32.1 and H24.9), their effect of suppressing $x$ remains consistent across different $k$ values. For other heads (*e.g.*, H39.7, H39.12, H25.13), their effect changes accordingly with $k$.

**Causal Effect of FI heads in Other Models.** Our previous analysis focuses on the causal effect of FI heads in Gemma-2 (9B). To further demonstrate the universality of our results across models, we repeat the initial validation (head ablation) experiments (§4; Fig. 4) with Llama-3 (8B), Mistral-v0.1 (7B), and Llama-2 (7B). Due to the different tokenization methods of these models, we use addition in the range [0,9] for Mistral-v0.1 (7B) and Llama-2 (7B); [0,999] for Llama-3 (8B). We visualize the results in Fig. 27. Similar to the observations in Fig. 4, the models achieve non-trivial performance on off-by-one addition, but switch back to perform standard addition when FI heads were ablated.

In addition to the initial validation (head ablation) experiments, we also repeated the further validation (causal analysis) with these three additional models. The results were visualized in Fig. 28–30. Our claims made with Gemma-2 (9B) hold across these three models: each FI head sends out a

distinct signal, and their aggregated effect implements the +1 function. Please refer to the captions of Fig. 28–30 for detailed discussions on the role of each FI head.

One mixed result we have is that in `Llama-2 (7B)`, the oldest model among the four we investigated, the function induced by FI heads is closer to is closer to $f(x) = x + 2$ than $f(x) = x + 1$. This suggests that `Llama-2`'s FI heads may not be fully formed yet, which in turn explains `Llama-2`'s weaker performance on off-by-one addition (6% accuracy on the range [0,9]).

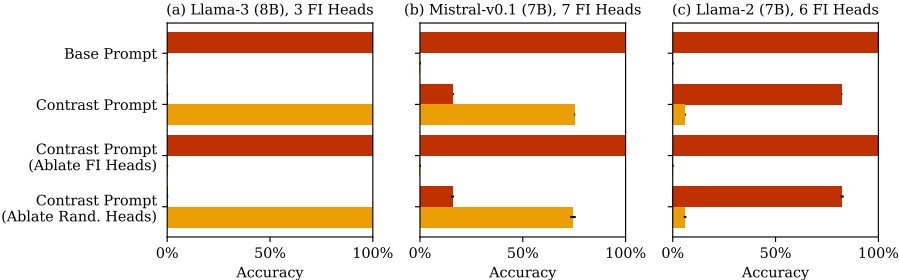

Figure 27: **Head Ablation Experiments, FI Heads, Three Additional Models.** In the random head ablation experiments, the number of ablated heads is equivalent to the number of FI heads identified in the model. The findings are consistent with those reported with Gemma-2 (9B) in Fig. 4.

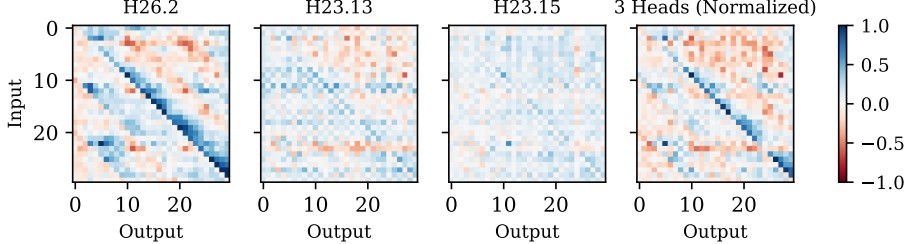

Figure 28: **Individual and Overall Effect of FI Heads in Off-by-$k$ Addition, $k = 1$, `Llama-3` (8B).** While `Llama-3 (8B)` uses [0,999] tokenization, we visualize results in the range of [0,29] for readability. H26.2 promotes $x + 1$ and sometimes $x + 2$; H23.13 and H23.15 promotes $x - 1$ and $x + 1$. They collectively contribute to promoting $x + 1$ as the output.

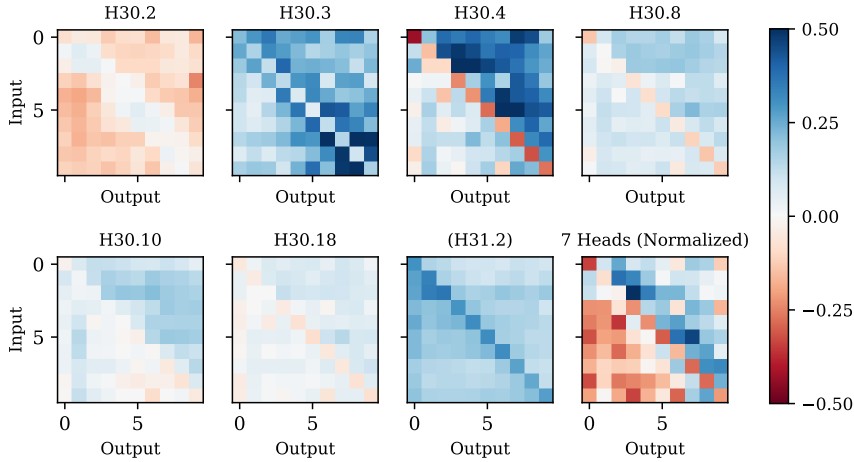

Figure 29: **Individual and Overall Effect of FI Heads in Off-by-$k$ Addition, $k = 1$, `Mistral-v0.1` (7B).** H30.3 promotes $x - 1$ and $x + 1$; H30.4 and H30.10 promote digits larger than $x$; H30.8 and H31.18 suppresses $x$. They collectively contribute to a function that's close to $f(x) = x + 1$.

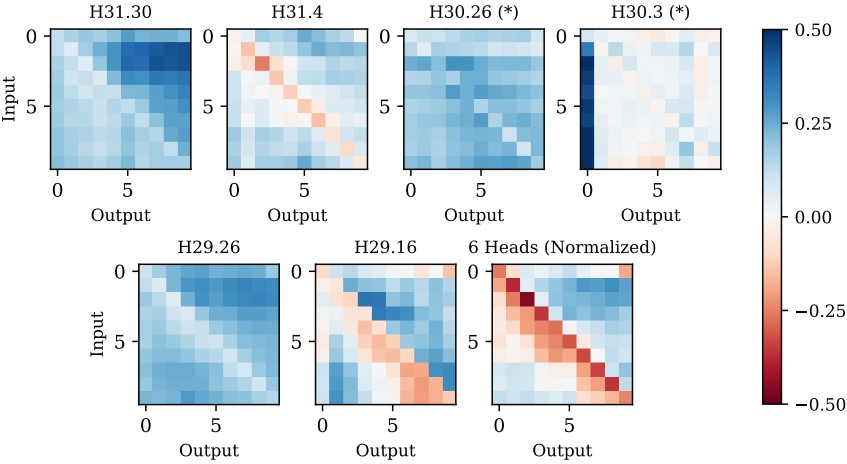

Figure 30: **Individual and Overall Effect of FI Heads in Off-by-$k$ Addition, $k = 1$, Llama-2 (7B).** H31.30 promotes numbers greater than $x$, H31.4 suppresses $x$, H30.26 suppresses $x$, H30.3 suppresses $x$ and promotes $x + 1$, H29.26 and H31.30 suppresses $x$, H29.16 suppresses $x - 1$, $x - 2$ and promotes $x + 1$. Their combined effect approximates $f(x) = x + 1$, though it leans toward $f(x) = x + 2$, which may account for Llama-2's weaker performance on this task. (*) Effects of H30.26 and H30.3 are rescaled to make the patterns more readable.

### E.2 Consolidation Heads

We repeat our function vector style analysis from §4, but this time use the consolidation heads as the subject. Concretely, we patch the outputs of these heads from the last token residual stream in off-by-one addition (*e.g.*, "1+1=3\n2+2=5\n3+3=?") to the naive prompts (*e.g.*, "2=2\n3=?"). We report the effect of this intervention on the output logits in Fig. 31.

We observe that three of these heads (H41.4, H40.11, H40.12) are suppressing answers other than $x$, and one head (H41.5) is promoting answers other than $x$. Their aggregated effect leads to promoting $x$ and suppressing $x+1$, which counters the effect brought by FI heads discussed in Fig. 5.

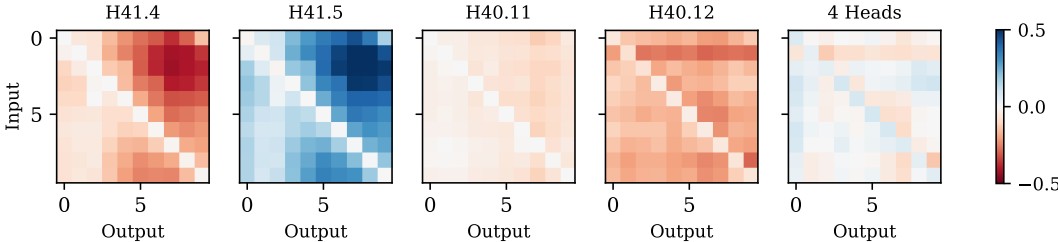

Figure 31: **Causal Effect of Consolidation Heads.** The aggregated effect of consolidation heads counter the effect of FI heads by promoting $x$ and suppressing $x+1$.

**Discussion.** We name these heads as "consolidation heads" based on three observations: **(1)** they appear in the final two layers; **(2)** they attend exclusively to the current token and the <bos> token, suggesting that they mainly process information locally at the current token; **(3)** our causal analysis in Fig. 31 shows that some of them are promoting 3+3=6 and some are promoting 3+3=7 in off-by-one addition, suggesting that they are weighing the two possible outputs collaboratively.

Despite these observations, our understanding on the exact role of these heads, and why they emerge, remain limited. We believe it relates to the broader phenomenon of "negative" behavior in language models, which has been noted as a challenge for current interpretability methods (Sharkey et al., 2025). We hope future work will present a finer-grained interpretation of these heads.

### E.3 Previous Token (PT) Heads

**Head Ablation Experiments.** To validate the role of previous token heads, we first repeat the head ablation experiments in Fig. 4, but we ablated previous token heads instead. We consider both off-by-one addition and off-by-two addition, and use 16-shots in the prompt. We report the results in Fig. 32. Ablating the previous tokens heads almost completely restores the model's default behavior. This supports our hypothesis that these heads are critical for inducing the +1 function.

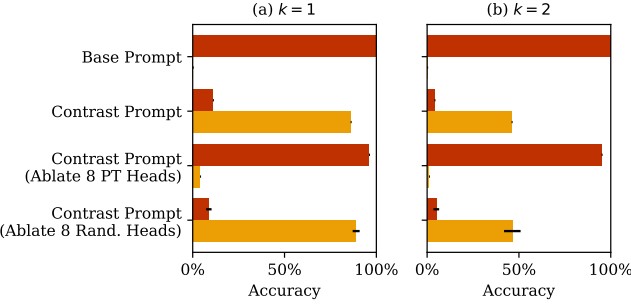

Figure 32: **Head Ablation Experiments, Previous Token Heads, `Gemma-2 (9B)`.**

**Causal Effect.** To further investigate the causal effect of previous token (PT) heads, we adapt our causal analysis method previously used for FI heads and consolidation heads. For example, consider the off-by-one addition prompt, "1+1=3\n2+2=5\n3+3=?", we extract the PT head outputs at the

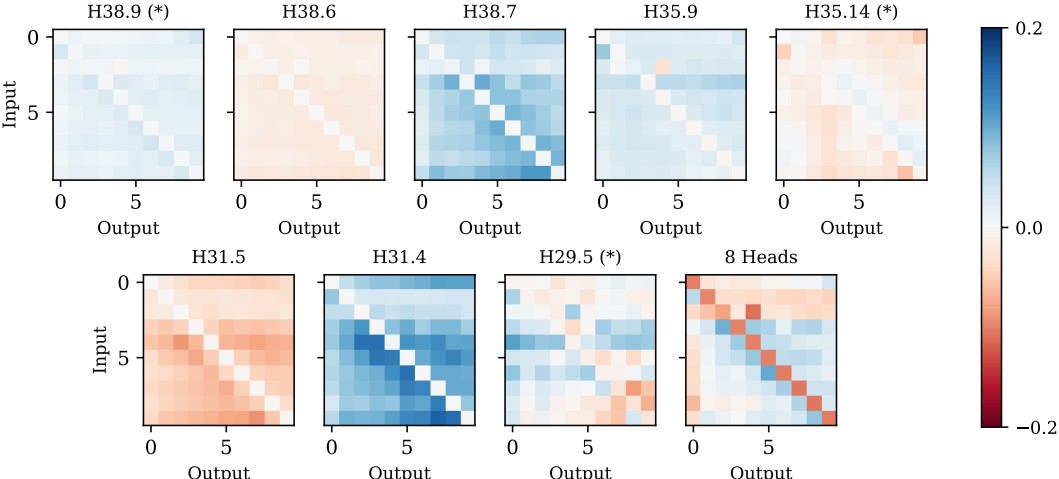

Figure 33: **Causal Effect of Previous Token Heads.** (*) Effects of H38.9, H35.14 and H29.5 are rescaled to [-0.02, 0.02] to make the patterns more readable.

tokens marked in brown, average the outputs over the two tokens (3 and 5), and add them to the forward pass of the naive prompt "2=2\n3=?", at the token 2 marked in brown. In our experiments, we scaled this explanatory example to 100 4-shot examples of off-by-one addition to extract the PT head outputs.

We report the causal effect of these PT head outputs in Fig. 33. Similar to the findings with FI heads, we find that each PT head conveys distinct information. For example, H38.7 promotes $x \pm 1$ and $x \pm 2$, H35.14 promotes $x + 1$ and suppresses $x - 1$. Collectively, these heads suppress $x$ and promote $x + 1$, which aligns with the hypothesized role of PT heads. Unexpectedly, these heads also promotes $x - 1$. We attribute this to the task shift from off-by-one addition and the naive prompt, and the token-level averaging operation we employ which may cause loss of information.

**Pairs of Heads with Countering Effect.** We notice that two pairs of PT head (H38.6/7 and H31.4/5) demonstrate opposing patterns, and they happen to be in the same group in group query attention. Similarly, two consolidation heads (H41.4/5; Fig. 31) have a similar countering effect. Hence we hypothesize that group query attention may help these heads develop countering or hedging behaviors.

## F    TASK GENERALIZATION

### F.1    TASKS AND DATA PREPARATION

In this section we describe the task pairs we used in §5 with more details.

**Off-by-$k$ Addition.** For experiments in the range of [0,9], we consider $k \in \{-2, -1, 1, 2\}$. For experiments in the range of [0,99] and [0,999], we consider $k \in \{-10, -9, ..., -1, 1, 2, ..., 10\}$. We have reported the results in Fig. 13-14, incorporating the range and offset information. We use 16 shots in the experiments in Fig. 6(a).

**Shifted Multiple-choice QA.** We focus on 6 subjects in the MMLU dataset (Hendrycks et al., 2021): high school government and politics, high school US history, US foreign policy, marketing, high school psychology, sociology. We downloaded the MMLU dataset from ⚫ hendrycks/test. We chose these subjects because Gemma-2 (9B) achieves 90% accuracy with 5 shots on them. For subjects where Gemma-2 (9B) achieves lower accuracies, tracking and analyzing performance on Shift-by-One MMLU becomes challenging, because the model could score points by random guessing. We use 16 shots in the experiments in Fig. 6(b), where the 16 shots combine "validation" and "dev" examples from the original MMLU dataset.

**Caesar Cipher.** We adopted a cyclic approach where "a" is considered the next character after "z". We also included both lower-case or upper-case examples, *e.g.*, "c -> d" and "C -> D" are both valid examples in ROT-1. We use 16 shots in the experiments in Fig. 6(c).

In the early stages of this work, we experimented with multi-character Caesar cipher. To prevent multiple characters from being tokenized as a single unit (*e.g.*, "ew" as one token in Gemma-2's tokenizer), we used a preceding whitespace (␣) before each character, formatting it as "␣e␣w" so that "␣e" and "␣w" became separate tokens. However, we ultimately focused on one-character Caesar cipher in the experiments because Gemma-2 (9B) has insufficient performance on the multi-character version. The tokenization-aware formatting was retained. The actual model input will be "␣c -> ␣d" for the example "c -> d".

**Base-$k$ Addition.** We sampled two-digit addition problems using a procedure similar to off-by-$k$ addition, with one additional constraint that the sum number $c$ has two digits in both base-10 and base-$k$. We use 32 shots in the experiments in Fig. 6(d). For the base-8 addition analysis in §5.2 and Table 3, examples for Case 1-3 were resampled.

## F.2 RESULTS

**Full Results using Different Offsets and Bases.** Previously in Fig. 6, we report results on representative cases, *e.g.*, $k = 2$ in off-by-$k$ addition, the subject of "high school government and politics" in shifted MMLU. In Fig. 34-36, we report results of the full list of offsets and subjects.

We observe that some of these task variants exceed Gemma-2 (9B)'s capabilities. For instance, Gemma-2 (9B) has notable performance on cipher when $k \in \{-2, -1, 1, 2, 3, 13\}$ but shows insufficient performance in other settings. Similarly, it only exhibits non-trivial performance on certain subjects of Shifted MMLU. However, when models do have non-trivial performance, we consistently see the involvement of the FI heads, evidenced by the decreased contrast accuracy after ablating them.

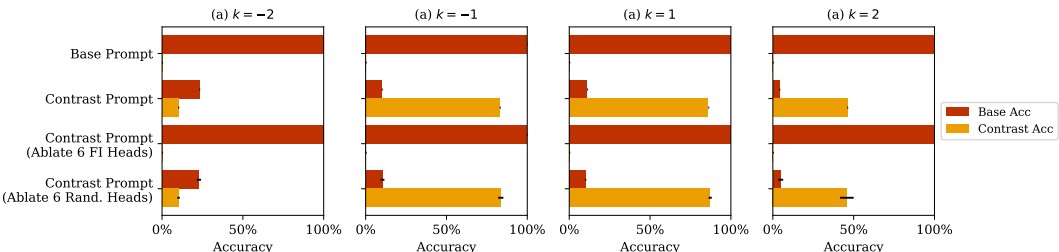

Figure 34: **Task Generalization with FI Heads, Off-by-$k$ Addition.** We consider addition in the range of [0,9] and $k \in \{-2, -1, 1, 2\}$.

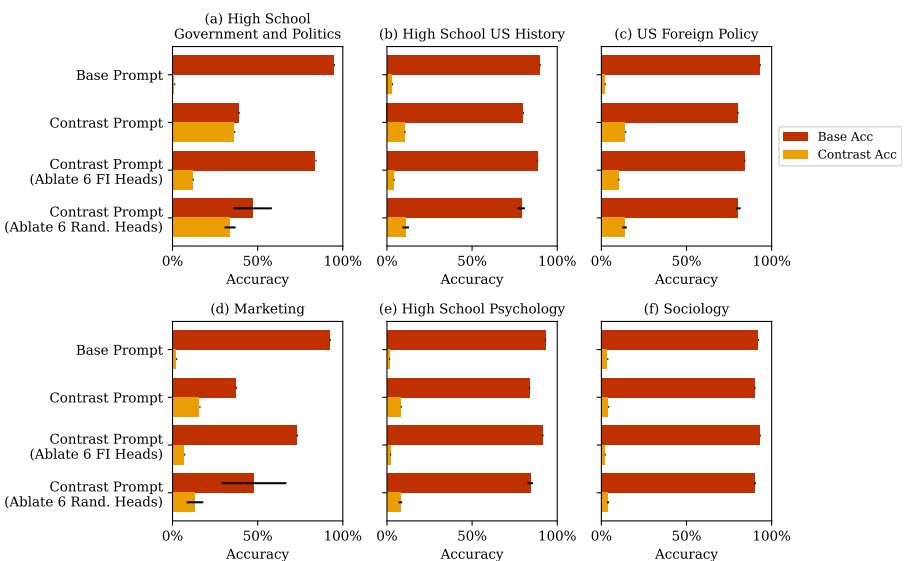

Figure 35: **Task Generalization with FI Heads, Shifted MMLU.**

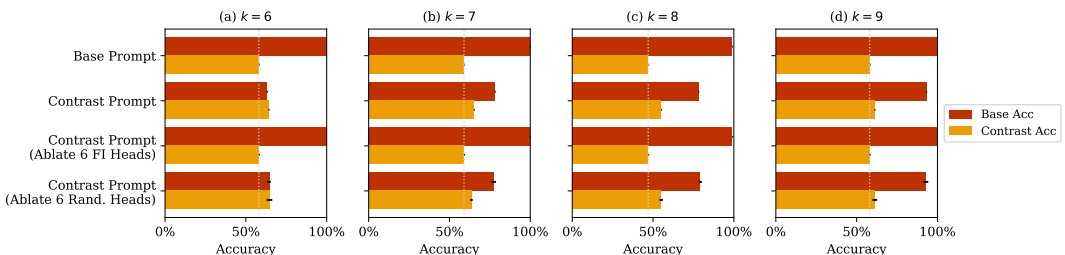

Figure 36: **Task Generalization with FI Heads, Base-$k$ Addition.** We consider $k \in \{6, 7, 8, 9\}$. The dashed lines represent the base prompt's contrast accuracy, emphasizing the delta in contrast accuracies between rows.

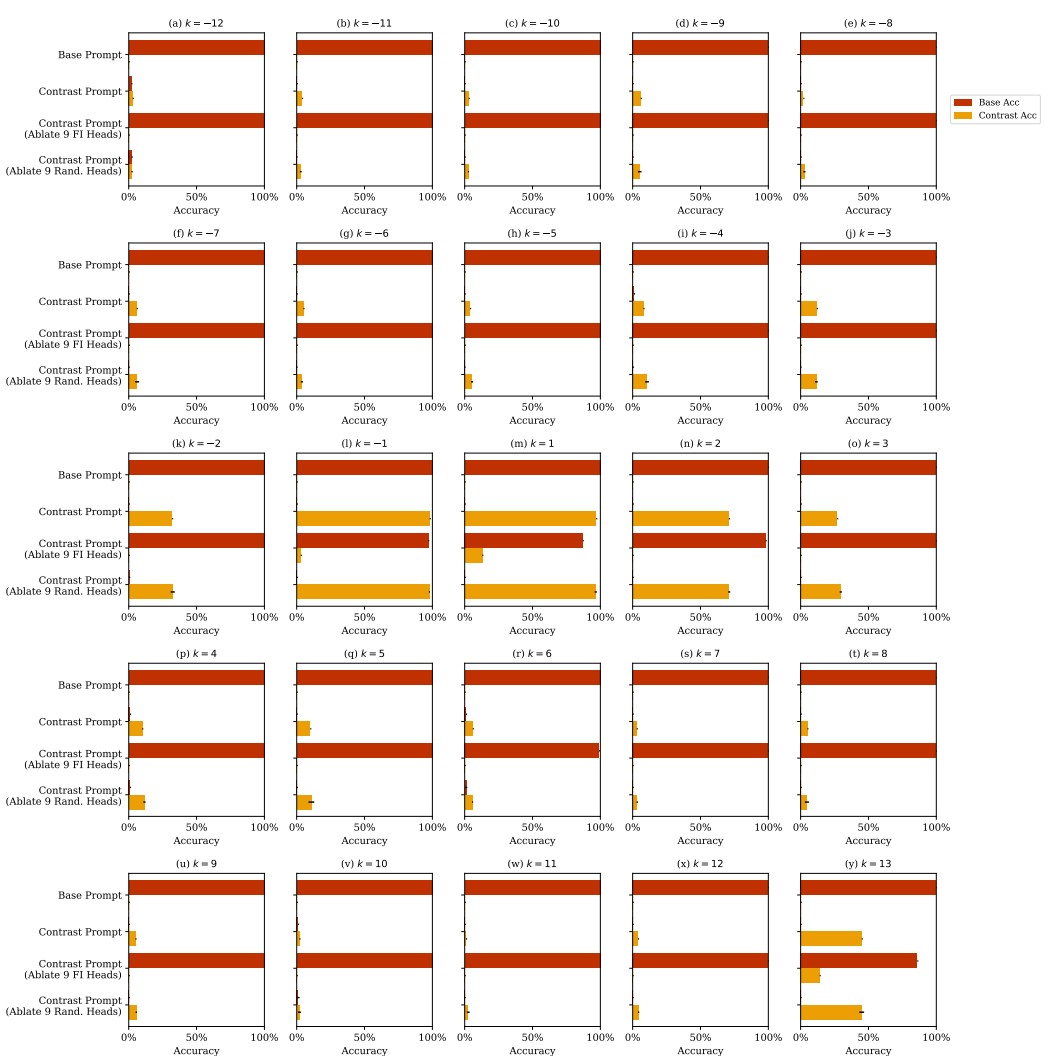

Figure 37: **Task Generalization with FI Heads, Caesar Cipher.** We consider $k \in \{-12, -11, ..., -1\}$ and $\{1, 2, ..., 13\}$. In this figure, we ablate 6 FI heads plus 3 additional FI heads (discussed in §4 and Fig. 5), yielding a clearer pattern than ablating 6 heads alone.

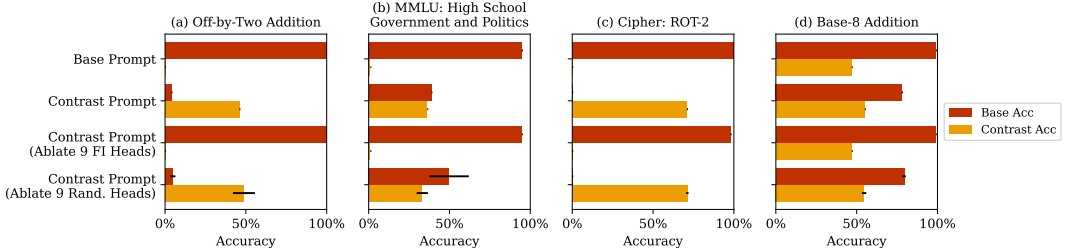

Figure 38: **Task Generalization with FI Heads, Ablating 9 FI Heads.** We repeat the experiment in Fig. 6, this time ablating three additional FI heads (H32.4, H28.6, H24.9) which showed a weaker effect ($1\% < |r| < 2\%$) during circuit discovery on off-by-one addition.

**Ablating three additional FI Heads.** Previously in Fig. 6, we ablate the 6 FI heads we identified in §3.2 by setting a threshold of $|r| > 2\%$. In §4 and Fig. 5 we showed that 3 additional FI heads with weaker effect ($1\% < |r| < 2\%$) also contribute meaningfully to off-by-one addition. Here we consider repeating the experiments on task generalization in Fig. 6 and ablating the 9 heads together. We report the results in Fig. 38.

We find that the 3 weaker heads contribute meaningfully to the Shifted MMLU, causing its contrast performance to drop to near 0% when all 9 heads are ablated (Fig. 38(b)), contrasting with 12% when 6 heads are ablated (Fig. 6(b)). We have a similar observation with Caesar Cipher ($k = 2$), where contrast accuracy drops to 0% in (Fig. 38(c)), contrasting with 36% when 6 heads are ablated (Fig. 6(c)). These observations suggest that the 3 heads may specialize in letters more than numbers. Understanding these detailed specializations will be an interesting direction for future work.

## G INSIGHTS FROM TRAINING TOY MODELS

Our work mainly focuses on interpreting off-the-shelf large language models, which is a common practice in works like Wang et al. (2023); Hendel et al. (2023); Todd et al. (2024). One alternative and promising research methodology is to train smaller transformer models from scratch, which allows precise control of the training data and more likely lets us isolate the circuit. This methodology is exemplified by works like Olsson et al. (2022); Nanda et al. (2023); Minegishi et al. (2025). We present a small preliminary study in this direction below.

Specifically, we trained a randomly-initialized, standard transformer model to perform addition (in the range of [0,99]) with 5-shot input examples. The transformer has $n_{layer} = 3, n_{head} = 4, d_{head} = 128, d_{ffn} = 512$. We use the Adam optimizer, with its weight decay set to 0.0001. We use an initial learning rate of 0.001, reduce the learning rate by half when the validation accuracy does not improve after 10 epochs, and stop training when the validation accuracy does not improve after 50 epochs.

We consider three different settings for the training data. We summarize the results in Table 6 and discuss our findings below.

| ↓ Trained on / → Test on | k=0 | k=1 | k=2 |
|---|---|---|---|
| k=0 | $98.3 \pm 1.0$ | $0.7 \pm 0.4$ | $0.0 \pm 0.0$ |
| k=0 and k=1 (50%/50%) | $53.7 \pm 5.4$ | $42.3 \pm 5.5$ | $1.7 \pm 3.2$ |
| k=0 and k=2 (50%/50%) | $16.0 \pm 1.4$ | $75.7 \pm 4.2$ | $10.6 \pm 3.8$ |

Table 6: **Results of Training Toy Transformer Models on Off-by-$k$ Addition.** We report mean and standard deviation over 5 runs.

- **Trained on standard addition (k=0)**: The model achieved an 98.3 accuracy on k=0, and near-zero accuracy on off-by-one addition (k=1).

- **Trained on k=0 and k=1, 50%/50% Mix**: The model achieved around 50% test accuracy on both k=0 and k=1, suggesting that it still could not infer the task in-context at the end of training. We tried several adjustments, such as increasing the number of layers or changing the mixing ratio of the training data, but none of these yielded improvements.

- **Trained on k=0 and k=2, 50%/50% Mix**: We explored whether this would enable the model to generalize to k=1. The model reached 16.0% accuracy on k=0, 10.6% on k=2, but surprisingly 75.7% on k=1. It appears the model averages the two training tasks: when trained on 1+1=2 and 1+1=4, it tends to output 1+1=3.

Overall, these results suggest that training a toy model to perform off-by-one addition is non-trivial. It likely requires specific changes to the data distribution or training curriculum. The search space is large and requires a separate study to address fully. We will leave this as future work.

## H   REPRODUCIBILITY

**Code Release.**   Our code and data are released at  INK-USC/function-induction.

**Frameworks.**   We primarily use the  transformer-lens (Nanda and Bloom, 2022) library for model inference and interpretability analysis. This library is built on the  transformers (Wolf et al., 2020) library. We have also used the  llm-transparency-tool (Ferrando and Voita, 2024; Tufanov et al., 2024) for early exploration.

**Hardware.**   All experiments were conducted with one NVIDIA RTX A6000 GPU (48GB). Path patching experiments involving 100 4-shot examples and iterating over all attention heads for a given target node will typically take 2 hours.

## I   LARGE LANGUAGE MODEL USAGE

After writing the initial draft, we refined it using language models to correct grammar, enhance clarity, and improve overall presentation. We also consulted language models for experiment implementation and matplotlib questions.

