# OpenReview forum: "Function Induction and Task Generalization: An Interpretability Study with Off-by-One Addition"
_ICLR.cc/2026/Conference — ICLR 2026 Poster_

### Official Review · Reviewer_Bkv2 · 2025-10-29

**Soundness:** 3
**Presentation:** 3
**Contribution:** 2
**Rating:** 4
**Confidence:** 3

**Summary:**

The paper introduces an in-context learning task called "off-by-one additio", where examples like `2+2=5`, `3+3=7`, and `4+4=?` require the model to infer the hidden “+1” rule. Using mechanistic interpretability, the authors identify a three-part circuit of attention heads:

1. Group 3 – Previous-Token heads: detect local patterns by attending from each answer to the preceding “=”, capturing the off-by-one discrepancy.
2. Group 2 – Function-Induction heads: attend from the test “=” to previous answers, encoding the learned “+1” transformation.
3. Group 1 – Consolidation heads: attend to the current token and `<bos>`, integrating signals to produce the final output.

Ablation studies show that Group 3 → Group 2 → Group 1 forms a causal chain implementing the “+1” rule.
The same mechanism generalizes to other tasks — off-by-*k* addition, shifted QA, Caesar ciphers, and base-8 arithmetic — revealing a reusable **function-induction circuit** within transformers.

**Strengths:**

1. The paper is clearly written and easy to follow, with a well-organized presentation of methods and results.
2. The evidence and ablation studies are convincing, providing solid empirical support for the proposed mechanisms.
3. The authors present an interesting and novel mechanistic explanation of in-context learning behavior in language models.

**Weaknesses:**

1. It is unclear how much of the observed flexibility of these attention heads originates from pretraining data. The paper assumes that the off-by-one addition task is novel, but it is plausible that similar unconventional arithmetic examples (e.g., math puzzles or riddles) appear in the pretraining corpus.
2. This raises two key questions:
   i. If the model were trained only on standard addition examples (e.g., `2+2=4`, `5+5=10`), would it still perform well on the off-by-one addition task?  (let's say we train them in an in context manner like the sequence of summations.)

   ii. If not, what aspect of the pretraining process enables these heads to adapt and represent such novel generalization?

3. These tasks appear quite simple; one would expect that even a relatively small transformer model, with significantly fewer layers, should be able to infer the in-context logic. With this in mind, I have the following questions:
i. If I replace the embeddings corresponding to these heads with zeros or random vectors, why can’t other heads—perhaps in later layers—compensate for that and still recover the in-context logic?
ii. What determines the specific role of each head? Is this role fixed during training?
iii. what happens if only use first few layers say 6 out of 24/36, can the model still recover the logic? Is the model uses other heads for doing so(if the answer is yes)?

Overall, I think the paper has a nice observation, but it is quite random to me that why certain heads are very important if we corrupt them other heads cannot compensate for that, and how much of all these in context learning capabilities is still linked to the pre-training phase.

**Questions:**

See Weaknesses

---

> ### Author Response · Authors · 2025-11-20
>
> Dear reviewer Bkv2,
>
> Thank you for your review and your encouraging words about our work! Regarding the weakness you mentioned, we would like to share some additional results and clarifications. Please note that we have reorganized the order of your questions for a more consistent narrative.
>
> > Weakness 2.1: If the model were trained only on standard addition examples (e.g., 2+2=4, 5+5=10), would it still perform well on the off-by-one addition task? (let's say we train them in an in context manner like the sequence of summations.)
>
> * Thank you for this suggestion. In the past week, we trained transformers from scratch on 5-shot in-context standard addition (k=0) within the range [0, 99]. This enabled the model to achieve >98% accuracy on k=0 but <1% accuracy on off-by-one addition (k=1). To directly answer your question: training on standard addition alone does not enable off-by-one addition.
> * This result intrigued us to explore two new settings, and we would like to share these findings with you. (1) We trained the model on both k=0 and k=1 examples. This led to ~50% test accuracy on k=0 and ~50% on k=1. The model could not infer the task in-context. We tried several approaches to help the model learn both tasks together, such as increasing the number of layers or changing the mixing ratio of k=0 and k=1 in the training data. However, none of these made a difference. (2) We trained the model on k=0 and k=2 examples, hoping this would enable generalization to k=1. The model achieved 16% accuracy on k=0, 76% on k=1, and 11% on k=2. This suggests that the model struggles to distinguish between k=0 and k=2, which surprisingly yields high accuracy on k=1. In other words, when we train the model on 1+1=2 and 1+1=4, the model tends to generate 1+1=3.
> * These results will be included in a new appendix section titled "Insights from Training Small Transformers."
>
> > Weakness 1: It is unclear how much of the observed flexibility of these attention heads originates from pretraining data.
>
> > Weakness 2.2: How does function induction emerge during pre-training? What data drive this?
>
> * Thank you for raising this important point! Linking model behavior back to pre-training data is indeed a critical question. In fact, we were curious about this as well and attempted to explore it—we evaluated Pythia-7B/12B models, which have their pre-training data publicly released. However, the model performance on off-by-one addition was low (below 30% at 32-shot), which prevented us from conducting further investigation.
> * We agree that it is possible the mechanism we identified was acquired from tasks like puzzles or riddles. Validating this hypothesis would also require access to released pre-training data, which unfortunately we do not have for the models we studied. We will include a footnote in the first paragraph of Section 3 to mention this possible cause. [Update: This discussion has been relocated to Section 7 in the revised version, where we felt it would be more appropriately situated.]
> * An ideal investigation of these questions would require suitable pre-training artifacts and significant infrastructure work, which is beyond the scope of this paper. We will add a discussion of future work in the conclusion section to invite future efforts on this important question.

---

> ### Author Response · Authors · 2025-11-20
>
> > Weakness 3.1 - “If I replace the embeddings corresponding to these heads with zeros or random vectors, why can’t other heads—perhaps in later layers—compensate for that and still recover the in-context logic?”
>
> * While compensation behavior is possible, we indeed did not observe it in our experiments. The complete mechanism we identified operates in the late layers of the model (Layers 25-41 in a 42-layer model, where Layer 41 is the final layer). We hypothesize that the remaining heads in these final layers likely serve other roles (e.g., processing non-algorithmic tasks) and that the model has not developed redundant backups for this +1 function.
>
> > Weakness 3.2 - “What determines the specific role of each head? Is this role fixed during training?”
>
> * The first question remains an open puzzle for the interpretability community. Pre-training on massive text corpora allows these heads to converge to their learned parameters, and various aspects of the training process, including data distribution, data order, optimization techniques, and initialization, could all contribute to determining each head's role. While we share your curiosity about this question, a comprehensive investigation is beyond the scope of our work.
> * Regarding the second question, a head's role can indeed change during training. For example, in the paper shared by Reviewer CMXz, attention heads' roles evolve through three distinct phases during training (see Figure 2 of https://openreview.net/forum?id=LNMfzv8TNb).
>
> > Weakness 3.3 - “What happens if you only use the first few layers, say 6 out of 24/36, can the model still recover the logic?”
>
> * We used a method called “logit lens” in Appendix D.4.1, which we believe addresses your question. The model cannot recover the logic using only early layers.
> * The model has 42 layers in total. According to Figure 18(b), when we decode from the first X layers (for all X <= 40), the model rates 3+3=6 as more likely than 3+3=7, meaning that it does not fully recover the +1 logic.
>
> > Weakness 3 - “One would expect that even a relatively small transformer model, with significantly fewer layers, should be able to infer the in-context logic.”
>
> * To clarify, this paper focuses on interpreting off-the-shelf, general-purpose language models such as Gemma-2 (9B) and Llama-3 (8B). General-purpose models must allocate their layers and attention heads across many capabilities, including question answering, creative writing, reasoning, etc. Therefore, we believe it is reasonable that only models above a certain size threshold can develop this mechanism.
>
> ---
>
> Thank you again for your thorough review and for raising many interesting questions! We are currently working on revising the paper to incorporate our responses to Weaknesses 1, 2.1, and 2.2. We will upload an updated version of the paper within the next few days. In the meantime, please let us know if you have any follow-up questions or would like us to clarify any points further.
>
> Authors of Submission 13646

---

### Official Review · Reviewer_xswB · 2025-10-30

**Soundness:** 3
**Presentation:** 3
**Contribution:** 3
**Rating:** 6
**Confidence:** 3

**Summary:**

This paper investigates how large language models (LLMs) generalize to unseen tasks in-context by analyzing a counterfactual arithmetic task called off-by-one addition.
Using mechanistic interpretability techniques, particularly path patching, the authors identify a novel computational structure, the Function Induction (FI) mechanism, which generalizes the known “induction head” mechanism from token-level copying to function-level generalization.

**Strengths:**

- The off-by-one task is simple yet counterfactual, enabling precise mechanistic tracing. The experiments are carefully designed to isolate the function-induction process.

- The analysis spans six modern LLMs (Gemma-2, Llama-2/3, Mistral, Qwen-2.5, Phi-4), confirming the generality of findings.

- The introduction of Function Induction (FI) heads extends prior induction-head results to function-level abstraction, a conceptually and methodologically significant advance.

- The paper provides one of the most concrete demonstrations of compositional and reusable circuits in LLMs, aligning with recent interpretability goals.

**Weaknesses:**

- Ambiguous motivation:

In Line 41, the paper states that “our understanding is still limited, especially regarding more complex generalization scenarios involving unexpected elements or newly defined concepts in the task.” However, the actual experiments focus on a very simple and synthetic task (off-by-one addition). It is therefore difficult to claim that the study meaningfully addresses “more complex generalization scenarios.”

- Unclear mechanism of Group 1 (consolidation heads):

The discovery of the Function Induction (FI) heads is an excellent contribution, but the role of Group 1 heads, those that most directly influence model outputs, remains vague. These heads attend strongly to the first token (e.g., `<bos>`), yet the reason such attention is crucial is not sufficiently explained.

- Limited scope and real-world implications:

The paper’s experiments are all conducted on toy arithmetic or algorithmic tasks. While this design is justifiable for achieving clear mechanistic insights and is a well-accepted approach in the mechanistic interpretability community, the paper would be stronger if it discussed how the identified mechanisms might translate to more realistic or linguistically complex tasks.

Despite these limitations, I found this paper highly compelling overall. The experiments are exceptionally clear, the evidence is convincing across multiple models, and the identification of FI heads represents a meaningful step forward in mechanistic interpretability research.

**Questions:**

In Figure 2, Group 3 indeed behaves like a previous-token head, but it also shows strong attention to the very first token (e.g., `<bos>`). Why does it attend so strongly to the first token? What role does this attention play in the function induction process?

---

> ### Author Response · Authors · 2025-11-20
>
> Dear reviewer xswB,
>
> Thank you very much for your feedback! We are so excited that you feel "the paper provides one of the most concrete demonstrations of compositional and reusable circuits in LLMs." We find these structures remarkable when we first identified them, and we appreciate your comments. Regarding your concerns, please see our response below.
>
> > Weakness 1 - [The paper] focuses on a very simple and synthetic task. It is difficult to claim that the study meaningfully addresses “more complex generalization scenarios.” (Line 41)
>
> * By "more complex generalization scenarios," we intended to create a contrast with prior works ([Hendel et al., 2023](https://arxiv.org/abs/2310.15916); [Todd et al., 2024](https://arxiv.org/abs/2310.15213)), which interpret _single-step, mapping-style_ tasks such as Country-to-Capital and Antonym. The tasks in our work are _two-step, algorithmic_ tasks, which represent a more complex scenario relative to these prior studies. We acknowledge that our tasks are still synthetic and that extending this analysis to more realistic tasks is an important direction for future work.
> * We will revise this sentence to clarify that prior works focus on single-step, mapping-style tasks and that our work examines more complex multi-step reasoning relative to those.
>
> > Weakness 3. Limited scope and real-world implications. “The paper would be stronger if it discussed how the identified mechanisms might translate to more realistic or linguistically complex tasks.”
>
> * Thank you for raising this important concern. While our investigation focuses on one particular mechanism in a controlled setting, we believe the findings have broader implications for language model development. Additionally, we believe this mechanism may be related to various model belief and behavior phenomena that are being actively investigated by researchers. We will revise the paper to include a discussion of these implications in the conclusion section, covering three key areas:
> * __Evaluation__: Our findings show that accuracy-based evaluation does not tell the full story. In the case of base-8 addition, models may achieve non-trivial accuracy while relying on shortcuts or unintended algorithms. Complementing accuracy-based evaluation with interpretability analysis can help reveal the model's true capabilities and whether it has learned the intended reasoning process.
> * __Pre-training__: Our analysis reveals that models perform multi-step reasoning latently -- a remarkable emergent structure given that these models are trained end-to-end on next-token prediction. This observation could inform the design of pre-training data mixtures or curricula that enhance compositional reasoning. For example, it may be beneficial to train early layers on single-step tasks (e.g., standard addition) before exposing the full model to multi-step tasks (e.g., off-by-one addition), thereby encouraging the development of function induction mechanisms and their reuse.
> * __Model belief and behavior__: Recent work has identified concerning behaviors in language models, such as sycophancy ([Sharma et al., 2023](https://arxiv.org/abs/2310.13548)) and susceptibility to belief shifts ([Geng et al., 2025](https://arxiv.org/abs/2511.01805)), which impact their reliability in real-world applications. We hypothesize that these behaviors may share structural similarities with the function induction mechanism we identified. Specifically, models may induce "belief-modifying functions" from context that drive these behaviors. Investigating this connection represents an important direction for future research.

---

> > ### Author Response · Authors · 2025-11-20
> >
> > > Weakness 2. Unclear mechanism of Group 1
> > * Due to space constraints, part of our analysis of Group 1 (consolidation) heads was deferred to Appendix F.2. We performed causal analysis and found that these heads collectively promote X and suppress X+1, which counters the effect of Group 2 (function induction) heads to some extent. This suggests that Group 1 heads play a stabilizing or hedging role in the overall computation.
> > * Regarding their `<bos>` attending behavior, we will explain it in the second paragraph in our response to Question 1.
> >
> > > Question 1. Why does it (H38.7, one of the Group 3 heads) attend so strongly to the first token `<bos>`? What role does this attention play in the function induction process?
> >
> > * To clarify, only the previous-token attending behavior at the "=" token is causally relevant for our task. While this head processes information at other positions in the sequence, such information does not causally affect the model output. We apologize for any confusion. In response to your feedback and Reviewer CMXz's feedback, we will revise Figures 2(c),(d),(e) to highlight the positions that are causally relevant for our task.
> > * The `<bos>` attending behavior is an interesting observation. We believe this is a separate phenomenon from the function induction mechanism in our work. For example, [Ferrando et al. (2025)](https://arxiv.org/abs/2504.02732) showed that “almost 80% of the attention is concentrated on the `<bos>` token” in Llama-3 405B. A common interpretation in the literature is that attending to <bos> represents an approximate "no-op" operation ([Gu et al., 2025](https://openreview.net/forum?id=78Nn4QJTEN); [Ferrando et al., 2025](https://arxiv.org/abs/2504.02732); [Clark et al., 2019](https://aclanthology.org/W19-4828/)). Since attention weights must sum to 1 due to the softmax operation, the model learns to attend to `<bos>` when attention to other tokens is not needed in the current context.
> >
> > ---
> >
> > Thank you again for your thoughtful review. We are currently revising the paper to address Weaknesses 2 and 3, include our discussion in Question 1, and update Figure 2. We will upload a new version within the next few days. Please do not hesitate to reach out if you have any follow-up questions.
> >
> > Authors of Submission 13646

---

> > > ### Comment · Reviewer_xswB · 2025-11-25
> > >
> > > Thank you for the detailed and thoughtful response. I appreciate the clarifications and the additional analysis, especially regarding the FI mechanism and the roles of the different head groups.
> > >
> > > That said, some of my core concerns, particularly the motivation framing and the broader implications beyond the toy setting, remain only partially addressed. For this reason, I am keeping my original score.
> > >
> > > Overall, I still find the work compelling and leaning toward acceptance, and I look forward to seeing the revised version.

---

> > > > ### Author Response · Authors · 2025-11-26
> > > >
> > > > Thank you for reading our response and getting back to us!
> > > >
> > > > We would like to leave a quick note that we have uploaded a revised version of the paper, including (1) revising Line 41 to better clarify our motivation; (2) including a new paragraph on "Implications for LLM Development and Applications" in Page 10.
> > > >
> > > > Thank you again and we welcome any further feedback you may have.

---

### Official Review · Reviewer_nVof · 2025-10-31

**Soundness:** 3
**Presentation:** 3
**Contribution:** 3
**Rating:** 8
**Confidence:** 3

**Summary:**

This paper studies in-context learning through the lens of a non-typical logic task: off-by-one addition. The idea is that inputs are two number sums (3 + 4) and outputs are the sum plus one (8). They use empirical mechanistic interpretability techniques to try to understand how the LLM solves the problem. In their results, they find a more abstract version of the induction head, they show multiple heads solve different parts of the task, and they show the same mechanism works with other, similar tasks beyond off-by-one addition.

In more detail, the main results are as follows:

- Off-the-shelf LLMs solve the off-by-one addition task in context and perform better with more examples.
- Path patching can be used to identify which parameters are contributing to different parts of the task. Path patching works by taking activations from a base task (regular addition) and replacing them in a network prompted to do the contrast task (off-by-one addition). If, after the patch, the network solves the base task, this is evidence that the patched part of the weights were key to the contrast part of the task (the +1 operation, for example). In this paper, all patches are the weights for particular attention heads. They identify several heads (they call them function induction heads), for which indeed replacing weights of the contrast task with the base task leads the network to revert to base task behavior (100% accuracy on the base task, 0% accuracy on the contrast task).
- They explore several other tasks besides off-by-one addition, and show similar phenomena.

**Strengths:**

- The path patching results are quite compelling. It is very surprising to me that replacing the heads from the contrast task with those of the base task leads to any sensible behavior at all, much less reversion to correct base task performance.
- I appreciate that the authors presented several other tasks in section 5. The results are a bit more mixed on the other tasks, but are nonetheless consistent with the narrative presented in the paper.

**Weaknesses:**

- There was not enough description of the differences between the FI head and the FV head. All that is reported is that the heads appear at different layers. The paper should describe conceptual differences between the solution concepts. If there aren’t major conceptual differences, this is a concern. The fact that the heads appear at different layers could be explained by other factors, like different models.
- I would quibble with “Off-by-one addition is likely an unseen task to these language models and represents a novel challenge”. Model producers and benchmarks for in-context learning often include synthetic tasks that don’t correspond to typical learning tasks. I think it is likely safe to assume models are trained on this kind of thing as well.

**Questions:**

What is the FV head and how is it different than the FI head?

---

> ### Author Response · Authors · 2025-11-20
>
> Dear reviewer nVof,
>
> Thank you for your review and for the encouragement! We are glad to hear that you find our work compelling and our analysis in Section 5 useful. We will address your comments below.
>
> > Weakness 1 and Question 1: Relation between FV heads ([Todd et al., 2024](https://arxiv.org/abs/2310.15213)) and FI heads (ours).
> * Thank you for flagging this confusion. We realize the current draft does not provide sufficient background on the FV heads work. In brief, Todd et al. discovered that a small number of attention heads transport task representations to enable in-context learning. Their study focuses primarily on _single-step, mapping-style_ tasks such as antonyms (fast -> slow, big -> ?) or country-capital mappings (France -> Paris, Australia -> ?). This contrasts with the task of interest in our study, off-by-one addition (1+1=3, 2+2=?), which is a _two-step, algorithmic_ task where the induced function represents the second step of the task (e.g., +1) and relies on successful computation of the first step (e.g., 1+1=2). While both FV heads and FI heads transport task representations, FI heads operate in a more complex, multi-step setting.
> * Our work also introduces two important findings not presented in Todd et al., 2024: (1) We identified the consolidation heads and previous-token heads that work together with FI heads to complete the full computation; (2) We provide a finer-grained interpretation showing that individual FI heads each write out a fraction of the "+1" function, which aggregates to form the complete "+1" function vector.
> * Finally, we would like to clarify that when we claim FV heads and FI heads are disjoint (Line 306), the comparison is performed within the _same_ model, Llama-2 (7B). We repeated our circuit discovery procedure on Llama-2 (7B) to enable this head-to-head comparison, so the disjointness cannot be attributed to model differences. Please refer to Appendix D.2.4 for further details.
>
> > Weakness 2: Off-by-one addition could be included in the pre-training data.
> * This is a valid point. We are unable to verify this either way. We will revise the sentence to: "Off-by-one addition is likely unseen or highly underrepresented in the pre-training data. Yet as Fig. 1 shows, the models effectively induce the +1 operation through in-context learning."
>
> ---
>
> Thank you again for your review. We will be happy to answer any follow-up questions.
>
> Authors of Submission 13646

---

> > ### Comment · Reviewer_nVof · 2025-11-25
> >
> > Thanks for your responses. I have also read the other reviews. My evaluation of the paper remains positive. Careful clarifying discussion of the role of FV heads will improve the readibility and contribution of this work, as several reviewers noted.

---

> > > ### Author Response · Authors · 2025-11-26
> > >
> > > Thank you for getting back to us! We have revised line 41 in the introduction to provide more context on FV heads, and made small edits at lines 265 and 314 to strengthen the background and improve signposting throughout. Thanks again!

---

### Official Review · Reviewer_CMXz · 2025-11-02

**Soundness:** 2
**Presentation:** 2
**Contribution:** 2
**Rating:** 4
**Confidence:** 4

**Summary:**

This paper investigates the mechanistic underpinnings of task-level generalization in LLMs through in-context learning. The authors use the novel, counterfactual task of off-by-one addition as a probe to understand how models adapt to new rules. This task requires a two-step process: performing standard addition and then applying an unexpected +1 function. Using circuit-style interpretability techniques, primarily path patching on the Gemma-2-9B model, the authors trace the internal computations responsible for this behavior. The authors find the identification of a circuit that generalizes the well-known induction head mechanism from the token level to the function level. Instead of just copying a token, this mechanism induces and applies an entire arithmetic function (+1) learned from the in-context examples. The circuit is composed of three distinct groups of attention heads: Previous Token heads, Function Induction heads, and Consolidation heads. Moreover, the authors show that this function induction mechanism is not specific to off-by-one addition. They provide evidence through head ablation experiments that the same core mechanism is reused by the model to solve a diverse range of other tasks requiring in-context rule adaptation, including off-by-k addition, shifted multiple-choice QA, Caesar ciphers, and base-8 addition. This suggests it is a general, flexible, and composable capability for task generalization.

**Strengths:**

[S1] This paper is well written and easy to follow even if without sufficient prior knowledge on this domain.

[S2] The experiment and analysis seem to be simple yet have effective coverage and depth.

[S3] The paper demonstrates analyses in the tasks where function-induction heads appear, suggesting that the function-induction heads in LLMs might be responsible for the capability to identify the off-by-one function and other new functions in in-context learning.

**Weaknesses:**

[W1] It is unclear how these results could generalize well and be universal across different models. Most analyses focus on the Gemma2-9B model, with function-induction heads only identified in other models but not sufficiently validated (we can see the results of Mistral/Llama in Appendix D.2) as done in Gemma2.  Moreover, usually the lines of mechanistic interpretability research (esp. induction heads) have started with simple toy models (e.g., a few layer attention networks) to clearly point out the target circuits in the model, and then expanded the analysis to LLMs. In contrast, this work just began with LLMs, where we could not exactly figure out the circuits. I think we might not be able to have higher confidence on the conclusion than the previous works.

[W2] It's not clear what aspect of function induction heads could be attributed as a generalization of induction heads. While induction heads could have a clear explanation about the role of key/value/query in the attention, the counterparts in function induction heads are unclear.

[W3] From the attention analysis in Figure 2 and Appendix D.2, I think it is quite challenging to distinguish among Group-1/2/3 heads at a glance. They all looked similar. Also, the analysis in the main paper is only about Group-2 heads (Figure 5/7). It may be necessary to clearly explain how we can classify those heads into Group-1/2/3 in the main text.

[W4] Limitation is not discussed in the main paper. Because the mechanistic interpretability may not perfectly explain how LLMs work in practice, the extensive discussion should be included.

[W5] While the results are interesting and validated to some extent, it is not really discussed how this finding could contribute to the progress of how LLMs are developed in practice. It is not clear what the takeaways from this analysis can be.

**Questions:**

Please also see **Weaknesses** section above.

[Q1] How should these findings suggest the way we evaluate, train, prompt, or otherwise use LLMs in practice? This paper may benefit more from an extensive discussion of how the function induction circuits on these tasks could be relevant for practical uses of LLMs in the real world.

[Q2] Could we expect/predict how these explanations transfer to larger models (more than 10B/100B/MoE, etc) well?

[Q3] Minegishi et al. (https://openreview.net/forum?id=LNMfzv8TNb) have studied the extension of induction heads similarly; the mechanism not only copying necessary tokens, but inferring the task from the context tokens and predicting the next-tokens corresponding to the query. Could you discuss the relevance and difference between them?

---

> ### Author Response · Authors · 2025-11-20
>
> Thank you for your thoughtful and detailed review! Please find our responses below. We are also preparing a revised paper that addresses your concerns and includes additional figures for clarification, which we will upload within the next few days.
>
> > Weakness 1 Part 1: “Function-induction heads only identified but not sufficiently validated.”
> * Thank you for raising the concern! To address this, we extended the experiments from Section 4 to include two additional models, Mistral-v0.1 (7B) and Llama-3 (8B), and the results corroborate our initial claims.
> * __Initial validation with head ablation.__ As reported in the table below, the results are consistent with the observations for Gemma-2 (9B) in Figure 4: ablating the identified FI heads flips the model's output from "off-by-one" addition back to standard addition.
>
> |                                         | Base Acc | Contrast Acc |
> |-----------------------------------------|----------|--------------|
> | **Mistral-v0.1 (7B)**                   |          |              |
> | Base Prompt                             | 100%     | 0%           |
> | Contrast Prompt                         | 16.0%    | 75%          |
> | Contrast Prompt (Ablate 7 FI heads)     | 100%     | 0%           |
> | Contrast Prompt (Ablate 7 random heads) | 15.8%    | 74.4%        |
> | **Llama-3 (8B)**                        |          |              |
> | Base Prompt                             | 100%     | 0%           |
> | Contrast Prompt                         | 0%       | 100%         |
> | Contrast Prompt (Ablate 3 FI heads)     | 100%     | 0%           |
> | Contrast Prompt (Ablate 3 random heads) | 0%       | 100%         |
>
> * __Further validation with causal analysis.__ We repeated the FI head causal analysis from Figure 5 on Llama-3 (8B) and Mistral-v0.1 (7B). The results confirm that these models perform off-by-one addition through similar mechanisms.
>   * In Llama-3 (8B), H26.2 promotes X+1 and sometimes X+2; H23.13 and H23.15 promotes X-1 and X+1. They collectively contribute to promoting X+1 as the output.
>   * In Mistral-v0.1 (7B), H30.3 promotes X-1 and X+1; H30.4 and H30.10 promote digits larger than X; H30.8 suppresses X. They collectively contribute to a function that’s close to X+1.
>   * We will add two new figures in the revised paper to visualize the causal effect of these heads.
>
> > Weakness 1 Part 2: "usually mechanistic interpretability research starts with simple toy models, … while this paper (directly) begins with LLMs”.
> * We believe both approaches are valuable and widely adopted in mechanistic interpretability research. For example, [Wang et al., 2023](https://arxiv.org/abs/2211.00593) (published at ICLR 2023) is a representative work that interprets LLMs directly. We chose to study large models directly because it allows us to investigate circuit reuse across a wider range of tasks, which is a main focus of this paper. However, we acknowledge this comes at the expense of circuit cleanliness.
> * That said, we agree that investigating off-by-one addition via toy models is valuable. Combining your suggestion with Reviewer Bkv2's, we conducted three preliminary experiments. In each, we trained a transformer from scratch to perform addition (range [0,99]) with 5-shot input examples.
>   * __Trained on standard addition (k=0):__ The model achieved >98% accuracy on k=0, but <1% accuracy on off-by-one addition (k=1).
>   * __Trained on k=0 and k=1:__ The model achieved ~50% test accuracy on both k=0 and k=1, but could not infer the task in-context. We tried several adjustments, such as increasing the number of layers or changing the mixing ratio of the training data, but none of these yielded improvements.
>   * __Trained on k=0 and k=2:__ We explored whether this would enable the model to generalize to k=1. The model reached 16% accuracy on k=0, 11% on k=2, but surprisingly 76% on k=1. It appears the model averages the tasks: when trained on 1+1=2 and 1+1=4, it tends to output 1+1=3.
> * Overall, these results suggest that training a toy model to perform off-by-one addition is non-trivial. It likely requires specific changes to the data distribution or training curriculum. This investigation space is large and would likely require a separate study to address fully. We will include the discussion above into the paper, leaving further investigation as future work.

---

> > ### Author Response · Authors · 2025-11-20
> >
> > > Weakness 2 Part 1: “what aspect of function induction heads could be attributed as a generalization of induction heads”.
> > * We can clarify this relationship using a concrete comparison:
> >
> > |            | Induction Heads                           | Function Induction      |
> > |------------|-------------------------------------------|-------------------------|
> > | Example    | Llama 0 was developed in 2021. Llama -> 0 | 1+1=3, 2+2=5, 3+3= -> 7 |
> > | Function f | f1 = “0”                                  | f2(x) = x+1             |
> >
> > * One way to view it is that induction heads (prior work) induce a _constant_ function and function induction heads (ours) induces a _linear_ function. This is what we meant by “function induction can be viewed as a generalization of induction heads”.
> >
> > > Weakness 2 Part 2: While induction heads could have a clear explanation about the role of key/value/query in the attention, the counterparts in function induction heads are unclear.
> > * Reusing the same examples above, we derive the following analogy, which we hope could explain what Q,K,V,O are responsible for in the function induction mechanism.
> > * Previous Token Head
> >   * Induction Heads: In the first occurrence of “Llama 0”, the query of “0” attends to the key of “Llama”. The value of “Llama” is transformed to represent “0 follows Llama” and stored at the position of “0”.
> >   * Function Induction: In the equation “2+2=5”, the query of “5” attends to the key of “=”, where a draft answer of “4” was stored. The value of “4” is transformed to the message “+1 is needed at =” at the position of “5”.
> > * (Function) Induction Heads
> >   * Induction Heads: In the second occurrence of “Llama”, the query of “Llama” attends to the key of “0”, and outputs “0” as the next token.
> >   * Function Induction: In the equation “3+3=”, the query of “=” attends to the key of “5” (which contains the “+1 is needed” message). It then writes the “+1 function” to the residual stream, causing the draft answer 6 to be revised to 7.
> > * We will also add a new version of Figure 8 by annotating what information is stored in the Q,K,V,O representations.
> >
> > > Weakness 3: “The three groups of heads (in Figure 2) all looked similar.”
> > * Thank you for flagging this concern!
> >   * For Group 1 heads, the pattern of interest is the main diagonal, especially at the “=” positions.
> >   * For Group 2 heads, the pattern of interest is 4 cells in the bottom row, and the rows every 6 rows above it, highlighting 3, 2, 1 cells respectively.
> >   * For Group 3 heads, the pattern of interest is the main diagonal shifted by one, corresponding to the previous token attending behavior. In particular, the green cell at row -2 and column -3, and the green cells every 6 rows above it are casually relevant to our task.
> > * We understand that it might be hard to map our textual description to the visual pattern here. We are working on a new version of Figure 2 to highlight the relevant cells, and will include it in the upcoming version.
> >
> > > Weakness 4: Limitation is not discussed in the main paper.
> > * Sorry about this. Now we are given a 10th page, we have moved the limitation section (originally Appendix A) to the main paper. Please see our upcoming version.

---

> > > ### Author Response · Authors · 2025-11-20
> > >
> > > > Weakness 5: It is not really discussed how this finding could contribute to the progress of how LLMs are developed in practice. It is not clear what the takeaways from this analysis can be.
> > >
> > > > Question 1: How should these findings suggest the way we evaluate, train, prompt, or otherwise use LLMs in practice?
> > >
> > > * Thank you for raising these questions! Although our work focuses on a specific task and mechanism, we believe the findings have broader implications for LLM development and application. We will revise the paper to include a discussion of these implications in the conclusion, covering three key areas:
> > > * __Evaluation:__ Accuracy-based evaluation does not tell the full story. In the case of base-8 addition, models may achieve high accuracy while relying on shortcuts or unintended algorithms. Complementing accuracy-based evaluation with interpretability analysis can help reveal the model's true capabilities and whether it has learned the intended reasoning process.
> > > * __Pre-training:__ Our analysis reveals that models perform multi-step reasoning latently -- a remarkable emergent structure given that these models are trained end-to-end on next-token prediction. This observation could inform the design of pre-training data mixtures or curricula that enhance compositional reasoning. For example, it may be beneficial to train early layers on single-step tasks (e.g., standard addition) before exposing the full model to multi-step tasks (e.g., off-by-one addition), thereby encouraging the development of function induction mechanisms and their reuse.
> > > * __Model belief and behavior:__ Recent work has identified concerning behaviors in language models, such as [sycophancy](https://arxiv.org/abs/2310.13548) and [susceptibility to belief shifts](https://arxiv.org/abs/2511.01805), which impact their reliability in real-world applications. We hypothesize that these behaviors may share structural similarities with the function induction mechanism we identified. Specifically, models may induce "belief-modifying functions" from context that drive these behaviors. Investigating this connection represents an important direction for future research.
> > >
> > > > Question 2: How will our findings transfer to larger models (more than 10B/100B/MoE, etc).
> > >
> > > * Larger models will have increased depth (number of layers) and breadth (number of heads per layer, and its hidden dimension). Along these two axes, we can make several speculations. Let $z=f(g(x))$ denote the task of off-by-one addition, where $y=g(x)$ represents standard addition, and $z=f(y)$ represents the +1 function.
> > > * __Impact of depth.__ Deeper models may be able to chain together more steps of function induction, e.g., $z=f_1(f_2(f_3(g(x)))$; Also, deeper models may be able to chain the +1 function to more complex $g(x)$ functions. Such $g(x)$ might have exhausted the layer capacity in shallower models.
> > > * __Impact of breadth.__ With more heads in each layer, the model may develop a broader library of inducible functions, e.g., $f_1(x)=x^2$, $f_2(x)=\sqrt{x}$.
> > >
> > > > Question 3: Relevance and difference with Minegishi et al. (https://openreview.net/forum?id=LNMfzv8TNb)
> > > * Thanks for pointing us to this work. In terms of the design, both work studies how transformer models perform in-context learning, using a group of related tasks. Minegishi et al. uses a classification style task while we focus on algorithmic tasks. Additionally, Minegishi et al. train small transformer models from scratch (enabling them to analyze circuit emergence), whereas we interpret large off-the-shelf models and focus on a static checkpoint.
> > > * In terms of the findings. Minegishi et al. identified a two-head circuit whose attention patterns and connections align with our "previous token head" and "function induction head." Both works find that models tend to perform a single task via multiple parallel paths. Both works also observe that models can adopt shortcut solutions for certain tasks based on existing mechanisms. Our unique finding is that the mechanism we identified operates in two-step tasks, showing that models can perform latent multi-step reasoning with it. Furthermore, we find that this mechanism is reused across many other tasks.
> > > * Thank you again for suggesting this interesting work! We will add a new paragraph on induction heads to the related work section. This paragraph will include our earlier explanation of why function induction is a generalization of the original induction heads, as well as the discussion above regarding Minegishi et al.
> > >
> > > ---
> > >
> > > Thank you again for your review! These questions encouraged us to think deeply and improve our paper. Additionally, your literature suggestion helped us address a question raised by Reviewer Bkv2. We will upload an updated version of the paper soon. Please let us know if you have any further questions in the meantime.
> > >
> > > Authors of Submission 13646

---

> ### Comment · Reviewer_CMXz · 2025-11-28
>
> Thank you for the comprehensive response, extensive discussions with related works, and the significant effort put into the additional experiments, particularly extending the analysis to other models (Llama-3, Mistral) and attempting the toy model training (Appendix G). I have reviewed the revised paper and your detailed replies. While I appreciate the inclusion of the new model results and the discussion on the Q/K/V analogy, I still have concerns regarding the nature of the "Function Induction" mechanism and its classification as a fundamental generalization of induction heads.
>
> 1. The Toy Model Discrepancy
>
> - I find the new results in Appendix G, "where small transformers failed to learn off-by-one addition in a way that replicates the "function induction" behavior found in LLMs", to be a bit negative signal with the following reason. Standard induction heads are known to emerge reliably in small 2-layer toy models (as shown in Olsson et al., 2022). If "Function Induction" is indeed a fundamental structural generalization of the induction head mechanism, one would expect it to be learnable in a controlled toy setting. The fact that the toy models either failed to generalize or resorted to averaging tasks (learning k=0 and k=2 to output k=1) suggests that the mechanism observed in LLMs might not be a clean, fundamental circuit, but rather a complex artifact of massive scale or specific pre-training distributions. This might weaken the claim that you have identified a reusable "primitive" of transformer computation.
>
> 2. The Q/K/V Mechanism Intuition
>
> - Thank you for the textual explanation of the Q/K/V roles. However, the analogy still feels a bit strained much. In a standard induction head, the value vector moves a specific token representation (Copy A -> A). In your proposed Function Induction mechanism, the value vector at the second step must transport a "function instruction" (e.g., "add 1") rather than a content token. The leap from "copying a token" to "copying an instruction to modify a future computation" is massive. I am not fully convinced that the attention heads are "inducing a function" in the mechanistic sense, rather than simply retrieving a vector that happens to bias the MLP or residual stream toward a specific region of the output space. The current evidence (patching) shows that it happens, but the how (the specific encoding of "+1" in the value vector) remains opaque compared to the clarity we have for standard induction heads.
>
> Additional Questions:
> - Toy Models: Do you believe the failure of the toy models to learn this mechanism is due to model size (lack of "capacity" to encode functions) or data distribution? If it is data distribution, does this imply that "Function Induction" is not an architectural capability but purely a memorized heuristic from specific pre-training data (e.g., math riddles), distinguishing it fundamentally from the architectural emergence of standard induction heads?
>
> - Cross-Model Robustness: In the new experiments for Mistral and Llama-3, did you observe the same cleanliness in the attention patterns as in Gemma-2? Often, patching results can look convincing in aggregate, but the specific attention heads might not show the distinct "tri-stage" pattern (PT -> FI -> Consolidation) as clearly as the primary model studied.
>
> - While the work is interesting, the inability to replicate the mechanism in simplified settings makes me worry that the "Function Induction" naming might be overclaiming the universality of the observed behavior.
>
>
> Again, I really appliciate the authors' significant efforts to revise your paper. Based on the reasons above, I still lean to the rejection, but because of the strong empirical experiments in several LLMs, I can be convinced even if the paper is accepted (that's why my rating is 4). Either way it would be great for the authors to elaborate or deal with the points I raised.

---

> > ### Author Response · Authors · 2025-11-30
> >
> > Thank you for your constructive feedback! Your engagement and efforts are greatly appreciated! We wish to address some misunderstandings we noticed in this round and provide additional clarification.
> >
> > __Clarifications on the naming of “function induction”__
> > * We used the phrase “function induction” because the study examines inductive reasoning in a language model at the function level, supported by substantial empirical evidence throughout the paper.
> > * By “a generalization of the induction head mechanism”—we should have been clearer.  We don’t mean that function induction is a generalized, primitive mechanism that subsumes standard induction heads. Rather, we intend to say that it’s a generalized, special case of inductive reasoning and it resembles the structure of induction heads.
> >
> > * We've already taken steps to avoid over-claiming in the current paper: (1) we included “off-by-one addition” in the title to narrow the scope; (2) we explicitly discussed our scope in the limitation section.
> > * You're right that we could be more precise in places. Below are some edits we are proposing:
> >   * [Abstract] “This mechanism resembles the structure of the induction head mechanism found in prior work and elevates it to a higher level of abstraction.” -> “operates at a higher level of abstraction.”
> >   * [Introduction] “This observation leads to our hypothesis of a function induction mechanism—a notable generalization of the induction head mechanism that transcends token-level pattern matching to operate at the function-level.” -> “—a generalized, special case of inductive reasoning that operates at function-level."
> >   * We believe these revisions more precisely communicate our findings without compromising the substance of our contribution. We'll also revise the full paper for similar places where clearer language would help.
> >
> > Thanks again for raising this concern and helping us to communicate our ideas more clearly.
> >
> > __The Q/K/V Mechanism Intuition__
> > * Thank you for the feedback. We realize that we should distinguish between two views on "function induction": (1) a behavioral or empirical view, where the model demonstrates the function-level inductive reasoning behavior, and we investigate which components in the model are responsible; (2) a more mechanistic view, where one would hope to understand exactly how the functions originate in the Q/K/V of the attention heads, as suggested by the reviewer (“inducing a function in the mechanistic sense”).
> > * Our paper makes progress on the first view with substantial empirical evidence. We appreciate the reviewer’s suggestion on diving into the second view, and we see it as important future work. We'll clarify this distinction in the paper to avoid any confusion about the scope of what we're addressing.
> >
> > __The Toy Model Discrepancy + Discussion on Model Heuristic__
> > > Do you believe the failure of the toy models to learn this mechanism is due to model size (lack of "capacity" to encode functions) or data distribution?
> > * We've experimented with increasing model capacity by adding layers, but this doesn't improve accuracy. So we believe it's primarily a data distribution issue.
> > * Looking at our toy model results, the behavior seems reasonable given the composition and the symmetry of the training data. We believe the models are responding reasonably to what they are trained on.
> >
> > > If it is data distribution, does this imply that "Function Induction" is not an architectural capability but purely a memorized heuristic from specific pre-training data (e.g., math riddles), distinguishing it fundamentally from the architectural emergence of standard induction heads?
> > * This question gets at a key issue. We think "induction" exists at multiple levels: (1) the function of +1 as a “memorized heuristic”; (2) a group of functions (+1, +k, letter shift) as a more generalized “heuristic”; (3) a broader class of functions under some complexity constraint.
> > * We would argue that any one of these represents function-level inductive reasoning in the empirical view, as the model has found and applied an appropriate function among many possible alternatives when facing an unseen scenario.
> > * We recognize that level 3 is what the mechanistic view would ideally find—true architectural emergence. Our current evidence supports up to level 2. We don’t intend to claim that models have achieved level 3, and we'll be explicit about this distinction in the revised paper.

---

> ### Author Response · Authors · 2025-11-30
>
> __Cross-Model Robustness__
> * The current version has included attention patterns from three additional language models in Figures 15-17 (appendix). In these models, we found the same chain of “PT -> FI -> Consolidation” heads.
> * For causally-relevant positions, the attention patterns align with those in the main model Gemma-2 (9B). There are model-specific variations in non-causally relevant positions, but we believe these variations don’t affect our core conclusions.
>
> ---
>
> Thank you again for your thoughtful comments! Your questions have really pushed us to clarify our thinking and improve our work. We're grateful for the time you invested in our work.
>
> Authors of Submission 13646

---

### Author Response · Authors · 2025-11-26
**General Response**

Dear reviewers and area chair,

Thank you all for your very thoughtful comments! We are grateful that reviewers found our work well-organized (CMXz, Bkv2), compelling (nVof), interesting (Bkv2), with solid empirical support (Bkv2) and coverage across multiple tasks (CMXz, nVof). We particularly appreciate reviewer xswB's characterization of the work, which captures what we hoped to contribute:

* [Presents] conceptually and methodologically significant advance;
* Provides one of the most concrete demonstrations of compositional and reusable circuits in LLMs.

---

We have carefully considered your comments and incorporated revisions into our paper accordingly. A revised version has been uploaded, with changes marked in blue. Below is a summary of the changes:

__Edits:__
* Line 41: Revised to provide context of prior works, and create a contrast with our work and highlight the advances we’ve made in terms of complexity. (Reviewer xswB)
* Line 121: Softened to acknowledge that off-by-one addition may appear in the model's training data, rather than ruling it out entirely. (Reviewer nVof)
* Figure 2, Panel (c)-(e): Highlighted causally relevant cells in pink. This change is also applied to Figures 15-17, which conduct the same analysis on three additional models. (Reviewer CMXz)

__New paragraphs in the main paper:__
* Section 6 (Related Work): Added a new paragraph on "Induction Heads in LMs" to provide context on prior work and discuss our contributions. (Reviewer CMXz)
* Section 7 (Conclusion): Added a new paragraph on “Implications for LLM Development and Applications”, covering discussion on evaluation, pre-training and model behavior. (Reviewer CMXz, xswB)
* Section 7 (Conclusion): Added a new paragraph on “Future Work”, discussing open research questions, including those raised by Reviewer Bkv2.
* Moved the limitation section from Appendix A to the main paper following the suggestion by Reviewer CMXz.

__New appendix content:__
* Appendix A, Figure 9: Added an annotated diagram of the induction head and function induction mechanism, illustrating the roles of query, key, value, and output components. (Reviewer CMXz)
* Appendix A: Added detailed discussion of similarities and differences with Minegishi et al., 2025. (Reviewer CMXz)
* Appendix C.3: Included discussion on the <bos>-attending behavior in language models, with a cross-reference footnote on page 4. (Reviewer xswB)
* Appendix E.1: Added a section on “Causal Effect of FI heads in Other Models”, providing initial and further validation experiments with three additional models. (Reviewer CMXz)
* Appendix G: Added a section on “Insights from Training Toy Transformers on Off-by-one Addition”, presenting a preliminary study training small transformers on standard and off-by-k addition tasks (Reviewer CMXz and Bkv2)

---

We welcome any further questions or concerns, and we're happy to discuss them with you! Thank you again for reviewing our work! Your feedback has helped us strengthen the paper, and we're grateful for the time and care you've invested in reviewing it.

Authors of Submission 13646

---

### Meta-Review · Area_Chair_2KXN · 2026-01-12

**Summary:**

This paper presents a mechanistic interpretability study of in-context task generalization in LLMs, using off-by-one addition as a probe task. Through path patching on Gemma-2-9B, the authors identify a three-component attention circuit—Previous Token, Function Induction, and Consolidation heads—that implements a function-level analog of classical induction heads, enabling the model to infer and apply novel arithmetic rules from context. Evidence suggests this mechanism is reused across diverse rule-adaptation tasks (e.g., Caesar ciphers, base-8 addition), pointing to a general, composable capability for task-level generalization. However, the paper omits discussion of limitations inherent to mechanistic interpretability and does not articulate practical implications for LLM development or training.

**Reviewer Scores:**

NA

---

### Decision · Program_Chairs · 2026-01-26

Accept (Poster)